# Molecular subtypes of ALS are associated with differences in patient prognosis

Jarrett Eshima [1], Samantha A. O'Connor [1], Ethan Marschall[1], NYGC ALS Consortium*, Robert Bowser [2,3], Christopher L. Plaisier [1] & Barbara S. Smith[1]✉

Amyotrophic Lateral Sclerosis (ALS) is a neurodegenerative disease with poorly understood clinical heterogeneity, underscored by significant differences in patient age at onset, symptom progression, therapeutic response, disease duration, and comorbidity presentation. We perform a patient stratification analysis to better understand the variability in ALS pathology, utilizing postmortem frontal and motor cortex transcriptomes derived from 208 patients. Building on the emerging role of transposable element (TE) expression in ALS, we consider locus-specific TEs as distinct molecular features during stratification. Here, we identify three unique molecular subtypes in this ALS cohort, with significant differences in patient survival. These results suggest independent disease mechanisms drive some of the clinical heterogeneity in ALS.

ALS is a heterogenous neurodegenerative disease defined by the progressive loss of motor neuron function, eventually leading to respiratory failure and death. Clinical diagnosis remains slow, hampered by an absence of disease-specific biomarkers, subjective scoring metrics, and presentation of symptoms that overlap with other motor neuron disorders early in the disease course[1,2]. The lack of diagnostic and prognostic biomarkers has led to the utilization of a patient classification system based on the site of symptom onset (lower, upper, and bulbar), which poorly predicts differences in patient pathology, survival, treatment responsiveness, and symptom progression[3,4]. As a consequence, the current lack of effective ALS treatments are directly tied to underlying patient heterogeneity. Recent efforts have been directed towards identifying the phenotypes and mechanisms driving clinical heterogeneity in neurodegeneration. In Alzheimer's patients, neuroimaging-derived subtypes demonstrated differences in clinical presentation, survival, age of onset, rate of progression, and age of death, providing critical new insight into disease heterogeneity[5]. Similarly, in the context of ALS, one group has recently developed a predictive model to stratify patients and inform prognosis, using patient-derived clinical information[6].

Current strategies to assess the molecular foundation of ALS heterogeneity have primarily applied '-omic' methodologies in combination with unsupervised clustering for disease subtype discovery[7–9]. Tam et al. established an important foundation for this work, using frontal and motor postmortem cortex transcriptomics to stratify a cohort of 77 ALS patients into three distinct subtypes[7]. They further demonstrate the direct interplay between TDP-43 and transposable elements using eCLIP-seq, providing key insight into the pathological role of transposable elements in ALS, given the near ubiquitous nature of TDP-43 cellular inclusions (-97%)[7,10,11]. We aimed to build upon this work by establishing a direct link between the ALS subtypes and clinical outcomes, such as survival and age of onset.

Here, we leveraged the large patient cohort in Prudencio et al.[10], NCBI Gene Expression Omnibus (GEO) accession GSE153960, to elucidate the hypothesized subtype-driven heterogeneity in ALS. Patient stratification analysis was performed using RNA-sequencing (RNA-seq) expression data from the frontal and motor cortex of 208 ALS patients, corresponding to 451 unique tissue samples. Transposable elements (TE) were quantified at the locus-specific level, which resulted in the redefinition of one ALS subtype. Three distinct molecular subtypes

[1]School of Biological and Health Systems Engineering, Arizona State University, Tempe, AZ 85287, USA. [2]Departments of Translational Neuroscience and Neurology, Barrow Neurological Institute, Phoenix, AZ 85013, USA. [3]St. Joseph's Hospital and Medical Center, Department of Neurobiology, Barrow Neurological Institute, St. Joseph's Hospital and Medical Center, Phoenix, AZ, USA. *A list of authors and their affiliations appears at the end of the paper. ✉e-mail: BarbaraSmith@asu.edu

were identified, with significant differences in survival, defined by i) glial activation (ALS-Glia), ii) oxidative stress and altered synaptic signaling (ALS-Ox), and iii) transcriptional dysregulation (ALS-TD). Importantly, these subtypes capture most of the existing disease mechanisms previously associated with ALS neurodegeneration[12]. In addition, some of the subtype-specific genes and transcripts identified in this study have not been previously associated with ALS, offering additional insight into disease pathologies and potential targets for diagnostic or personalized therapeutic development.

## Results

### Unsupervised clustering identifies three molecular subtypes in the frontal and motor cortices of ALS patients

To test the hypothesis that ALS patient clinical heterogeneity is driven by subtype-specific disease mechanisms, we first performed an unsupervised clustering analysis using 451 ALS postmortem cortex transcriptomes (Fig. S1; table S1; Supplementary Data 1). SQuIRE[13] was implemented to quantify transposable element expression with chromosomal locus specificity (Supplementary Data 2). TE features were filtered to ensure the retained transcripts had unique mapping reads and quantifiable expression in all ALS patient samples. Prior to

clustering, a variance stabilizing transformation was applied (Fig. S2) and the removal of sex-dependent genes was performed using DESeq2 differential expression[14]. Estimation of factorization rank was then performed in R (Supplementary Data 3), and a rank of 3 was chosen given the quality metrics (Fig. S3). Similar to the approach outlined by Prudencio et al.[10], we split the cohort by sequencing platform (HiSeq 2500 and NovaSeq 6000, Illumina, San Diego, CA), to account for substantial batch effects in gene expression due to the use of different sequencing instruments.

After filtering for the top 10,000 most variably expressed genes, we applied non-smooth non-negative matrix factorization (nsNMF)[15] to identify subgroups of ALS patients based on gene expression in the postmortem cortex. Three distinct patterns of gene expression were identified in both the NovaSeq and HiSeq ALS cohorts (Fig. 1a, f). In the NovaSeq cohort there was roughly a ratio of 3:1.4:1 observed for the ALS-Ox, ALS-TD, and ALS-Glia subtypes, respectively. The HiSeq cohort showed a similar proportion of ALS subtypes, with an approximate 3:1.9:1 ratio observed for the ALS-Ox, ALS-TD, and ALS-Glia subtypes, respectively. Principal component analysis (PCA) demonstrated the ability to separate the putative ALS subtypes into three distinct clusters when considering the first and

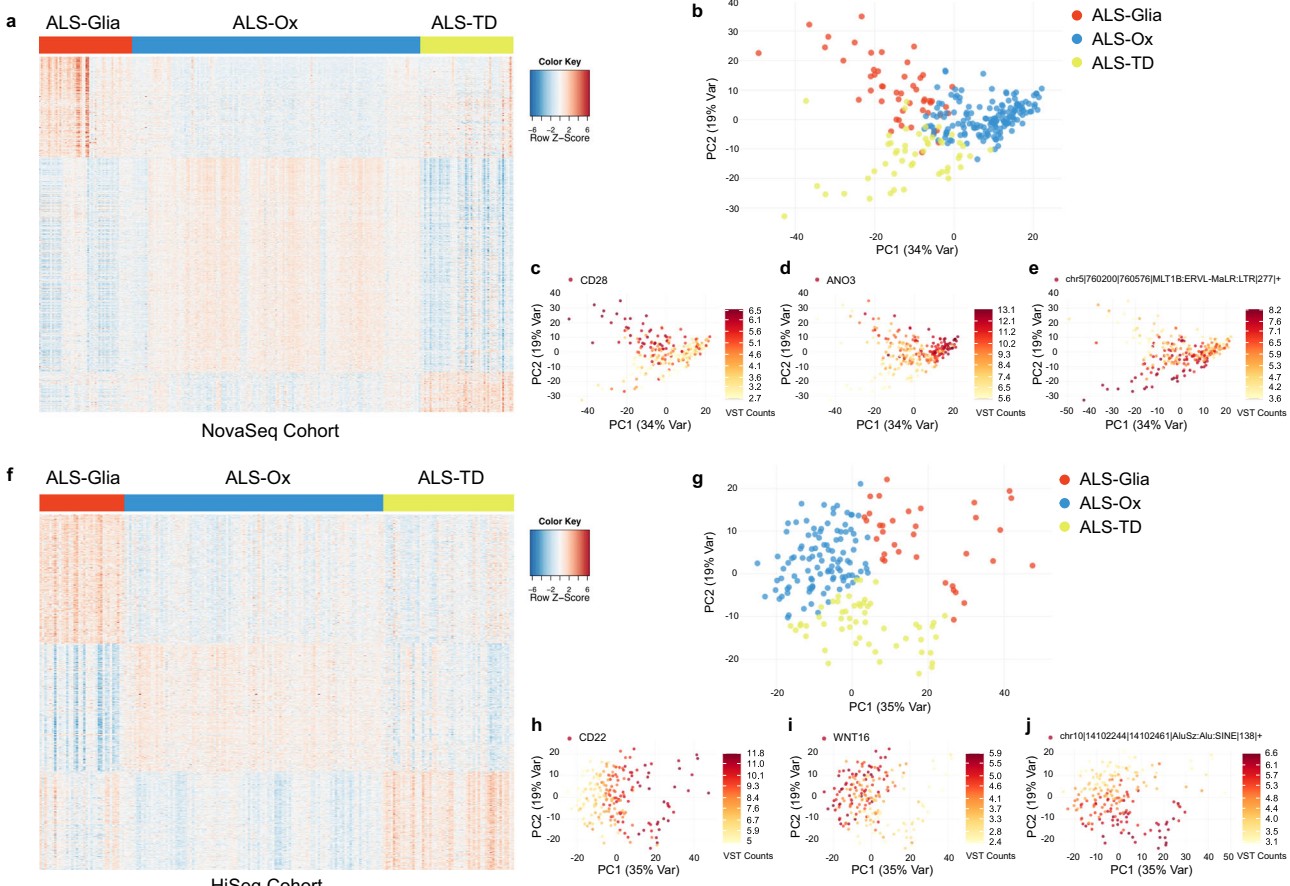

**Fig. 1 | Unsupervised clustering analysis with ALS postmortem cortex transcriptomes. a** Heatmap of 741 genes and transposable elements selected by SAKE[113] shows transcript overexpression in a subtype-specific fashion for the NovaSeq cohort (*n* = 255 biologically independent samples). Transcript counts are z-score normalized. **b** Principal component analysis shows three distinct clusters when considering the first two principal components. **c** Sample expression of *CD28* transcripts was plotted in the same PCA space, with elevated counts seen for the ALS-Glia subtype. A darker color corresponds to higher feature expression. **d** Expression of the *ANO3* gene shows specificity for the oxidative stress and altered

synaptic signaling subtype. **e** The ALS-TD subtype shows specific upregulation of transposable element chr5 | 760200 | 760576 | MLT1B:ERVL-MaLR:LTR | 277 | + compared to the other two subtypes. **f** Heatmap of 618 genes and TEs shows subtype-specific expression in the HiSeq cohort (*n* = 196 biologically independent samples). **g** PCA considering the HiSeq cohort shows three distinct clusters of ALS patient transcriptomes. **h** Elevated expression of *CD22* is seen in the activated glia subtype. **i** Subtype-specific expression of *WNT16* in the ALS-Ox subtype. **j** chr10 | 14102244 | 14102461 | AluSz:Alu:SINE | 138 | + is overexpressed in the ALS-TD subtype. Source data are provided as a Source Data file.

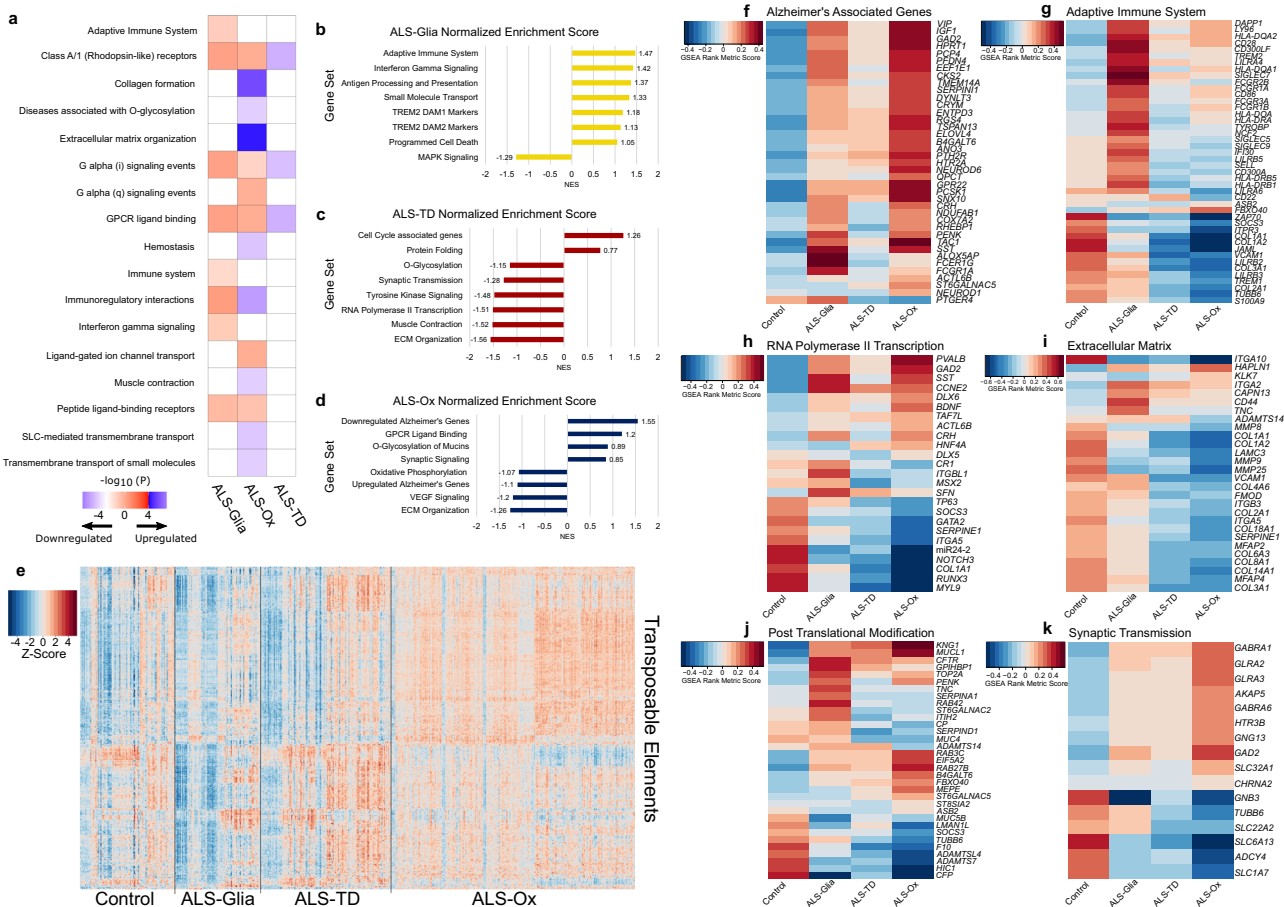

**Fig. 2 | Enrichment analysis identifies subtype-specific disease pathways.**
**a** Benjamini-Hochberg adjusted *p*-values, derived from a Fisher's exact test, are presented on the −log$_{10}$ scale. All presented pathways are significantly enriched in at least one subtype. Negative enrichment is encoded as the negative magnitude of the −log$_{10}$(adjusted *p*-value). *P*, Fisher's exact test, one-tailed, Benjamini-Hochberg method for multiple hypothesis test correction. **b**–**d** Gene sets enriched in each ALS subtype are presented along the Y-axis, with GSEA normalized enrichment score (NES) presented along the X-axis. **e** Heatmap of transposable element expression,

with 426 unique TEs and 544 biologically independent transcriptomes. Patient samples were plotted by subgroup, with the thin black lines denoting sample separation by subtype. TE count values were subject to VST, followed by z-score normalization, with red indicating elevated expression. **f**–**k** Pathways enriched specifically for one or more subtypes were generated using GSEA rank metric scores. Genes comprising each functional pathway are included, with subtype-specific gene enrichment scores encoded on a red-blue scale. Source data are provided as a Source Data file.

second principal components (Fig. 1b, g). Six transcripts associated with ALS-Glia, ALS-Ox, and ALS-TD were considered in the principal component space, and subtype-specific expression can be identified in both sequencing platform cohorts (Fig. 1c–e, h–j). Taken together, these results support the existence of three distinct patterns of gene and TE expression within the ALS postmortem cortex transcriptome.

### Gene set enrichment analysis reveals subtype-specific phenotypes

To elucidate subtype-specific molecular phenotypes, we performed hypergeometric enrichment analysis[16] (Fisher's exact test) and Gene Set Enrichment Analysis (GSEA)[17] using the top 1000 features from each sequencing platform cohort, leaving 1681 unique transcripts (Supplementary Data 4). Subtype-specific pathway enrichment was observed for each ALS subtype (Fig. 2a–d). In ALS-Glia samples, enrichment for immunological signaling and activation, genes implicated in a pro-neuroinflammatory microglia state in Alzheimer's (Disease-Associated Microglia, DAM)[18], and markers of neural cell death were observed (Fig. 2a, b, g). Transposable element expression was greatly reduced in ALS-Glia samples compared to the other two subtypes (Fig. 2e; Fig S4a, c).

Enrichment of the ALS-TD and ALS-Ox subtypes suggests some overlapping disease mechanisms, such as altered ECM maintenance and the influence of post-translational modification machinery (Fig. 2c, d, i, and j). Furthermore, while the ALS-Ox subtype had the strongest expression of the locus-specific TEs (Fig. 2e; Supplementary Data 5), the ALS-TD subtype showed elevated TE expression more often than the control groups and ALS-Glia subtype (Fig. S4a). To distinguish the ALS-TD subtype from ALS-Ox, we observe the unique downregulation of RNA polymerase II transcriptional genes (Fig. 2h). We utilized this evidence, along with univariate features considered later, to determine this ALS subgroup is defined by transcriptional dysregulation (TD), rather than TE expression[7].

In the ALS-Ox subtype we note distinct enrichment of Alzheimer's associated genes, but not genes previously associated with ALS or Parkinson's disease, which may reflect our stringent filtering during NMF score-based feature selection. We observed negative enrichment for genes involved in oxidative phosphorylation (Fig. 2d), and weak positive enrichment for synaptic signaling (Fig. 2d, k), when compared to the control cohort. It is worth noting that our subtype enrichment generally agrees with the findings reported in Tam et al.[7], despite the increased size of our patient cohort, although we elected not to include custom TE enrichment (additional details in Methods section,

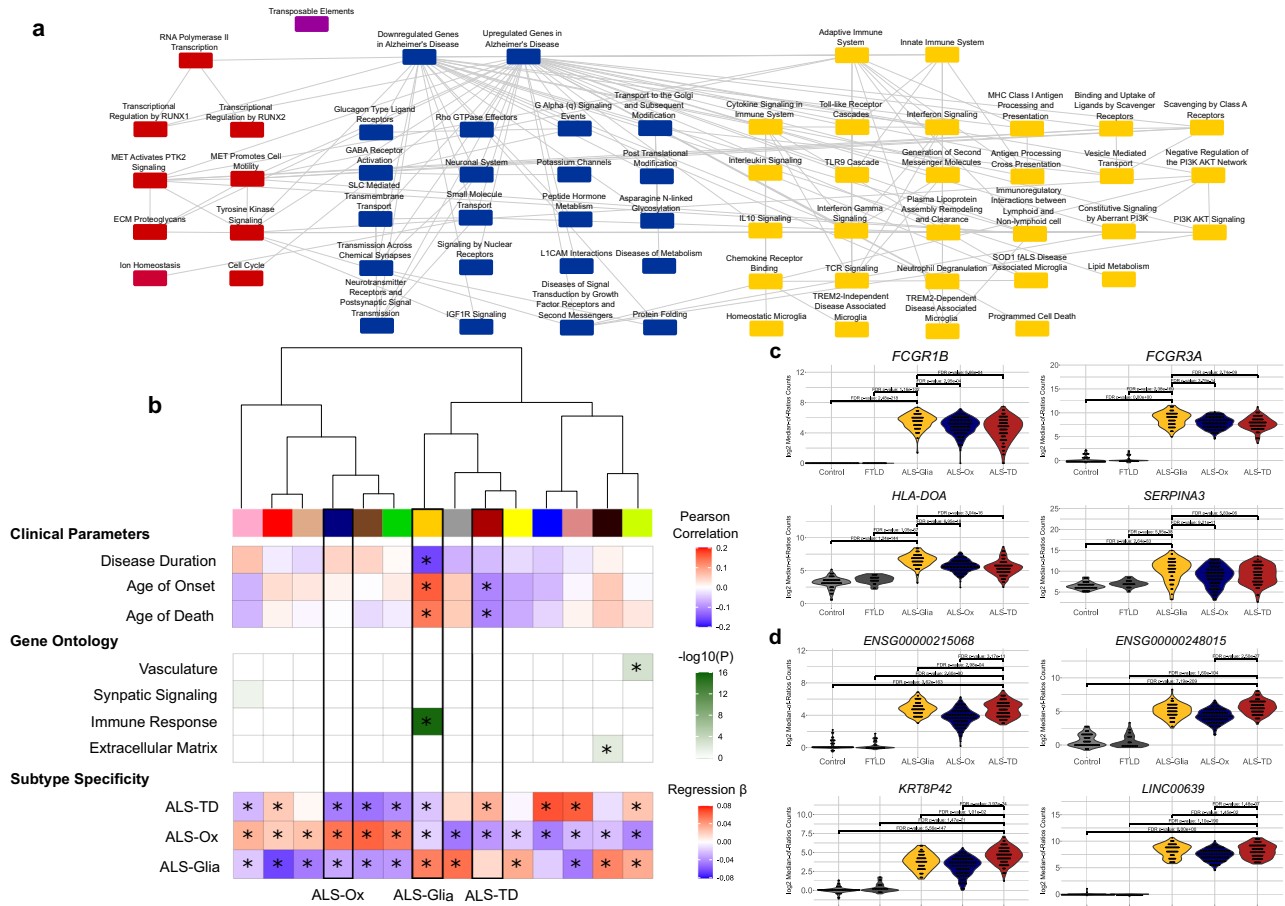

**Fig. 3 | Network construction elucidates subtype-specific disease pathways and eigengenes associated with ALS patient clinical outcomes. a** Network of pathways associated with the ALS cohort, color coded by subtype, with maroon indicating ALS-TD, navy denoting ALS-Ox, and gold signifying ALS-Glia. **b** WGCNA identifies gene subsets significantly correlated with ALS patient age of disease onset, age of death, and disease duration (univariate regression, two-tailed). Eigengene labels, moving left to right in the dendrogram, are: pink, red, tan, navy (ALS-Ox), brown, green, gold (ALS-Glia), gray, maroon (ALS-TD), yellow, blue, salmon, black, and green-yellow. Eigengenes were enriched for gene ontology and Bonferroni-adjusted *p*-values are shown (Fisher's exact test, one-sided). Subtype-specific expression of eigengenes was determined using dummy regression (two-tailed), with the β coefficient presented as a heatmap. A positive β coefficient denotes subtype upregulation of transcripts comprising the particular eigengene. Bonferroni-adjusted *p*-values less than 0.05 are denoted with *. **c** Univariate plots showing gene expression levels of four features (*FCGR1B, FCGR3A, HLA-DOA, SERPINA3*) in the gold eigengene – with evidence for ALS-Glia specificity. *P*, DESeq2[14] differential expression using the negative binomial distribution, two-tailed, false discovery rate (FDR) method for multiple hypothesis test correction. **d** ALS-TD specific expression of four representative features (*ENSG00000215068, ENSG00000248015, KRT8P42, LINC00639*) in the maroon eigengene. *P*, same as **c**. Source data are provided as a Source Data file.

Fig. S4; Supplementary Data 6) and some differences are observed for the ALS-Ox group. Given these results, we elected to maintain the ALS subtype naming conventions presented by Tam et al., where appropriate.

**Network development reveals gene subsets correlated with ALS disease duration, age of symptom onset, and age at death**

We constructed a network in Cytoscape[19] to facilitate the interpretation of subtype-specific pathway enrichment, utilizing the results from GSEA (Fig. 3a). Pathway nodes were manually color coded by subtype and edges denote overlapping genes between pathways. The transposable elements node is color coded purple to signify specificity for both the ALS-Ox and ALS-TD subtypes (Fig. S4a). Although informative, we sought to complement this analysis by identifying co-expressed gene sets (eigengenes) associated with patient clinical parameters, such as age of symptom onset, age of death, and disease duration using a weighted gene co-expression network analysis (WGCNA)[20]. This approach has proven successful in characterizing the functional and molecular differences that distinguish cellular composition and pathogenic processes in disease[21].

Our results indicate the maroon and gold eigengenes are significantly correlated with ALS clinical parameters (Fig. 3b). Expression of the maroon eigengene is seen to be negatively correlated with age of symptom onset and age at death. Conversely, the gold eigengene is seen to be positively correlated with age of onset and death, yet negatively correlated with disease duration (Fig. 3b). The observed relationship between the gold eigengene and patient clinical parameters indicates that elevated expression drives a later disease onset but a shorter survival duration. Identification of eigengene clusters based on the correlation of gene expression is shown as a dendrogram and heatmap plot (Fig. S5a) and tabulation of module membership is presented (Supplementary Data 7). The visualization of features comprising subtype-specific eigengenes[22] is presented in Fig. S5b–d.

Eigengenes were enriched for gene ontology (Fig. 3b; Supplementary Data 8), and the gold eigengene was seen to be strongly linked to the immune system ($p < 5 \times 10^{-16}$, Fisher exact test, one-tailed, Bonferroni-corrected). Importantly, we observed ALS-Glia specific overexpression for the majority of features included in the gold eigengene (Fig. 3c). The maroon eigengene – primarily composed of transposable elements, long non-coding RNA, pseudogenes, and

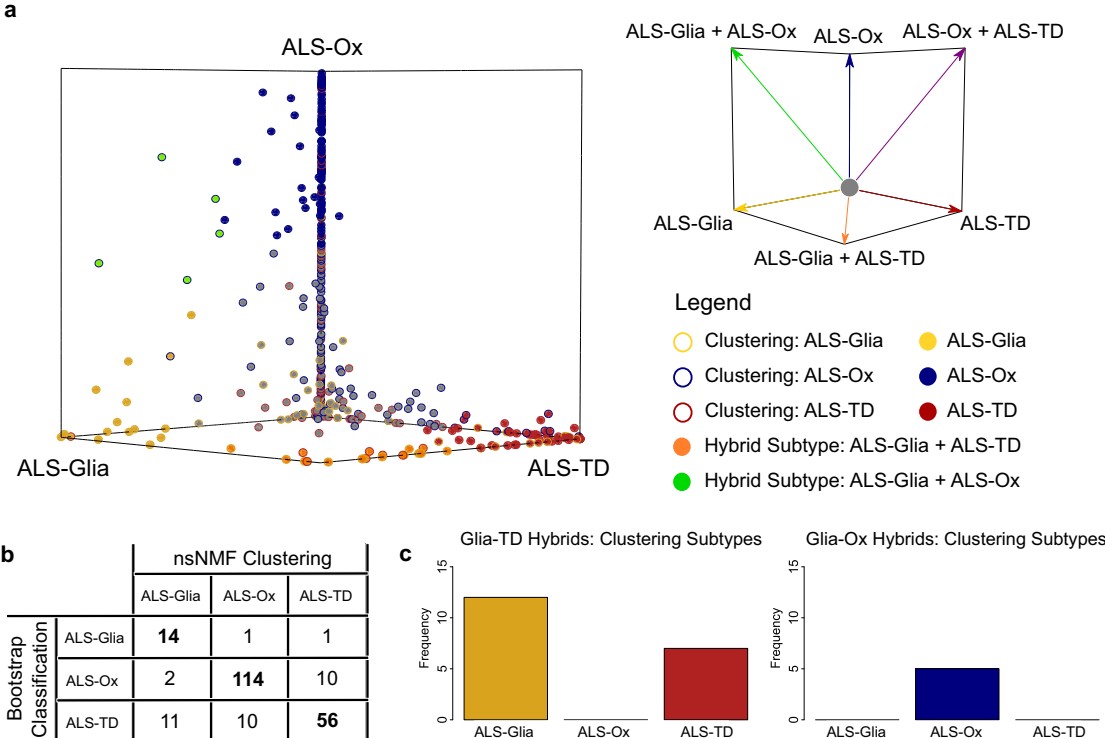

**Fig. 4 | Score-based classification uncovers hybrid subtype states in the ALS cohort. a** Subtype scoring was implemented with bootstrapping to assess the spectrum of disease phenotypes presented in ALS. Each point corresponds to a single transcriptome derived from the frontal or motor postmortem cortex, $n = 451$ biologically independent samples. Patient samples were initially placed at the origin, moved in the direction of the subtype axis for each round of bootstrapping that passed the subtype score threshold, and could only reach the vertex if the patient sample passed the threshold in all rounds of bootstrapping. Data points are filled according to the bootstrap-based subtype assignment and borders are included to denote the patient subtype obtained from unsupervised clustering. Transcriptomes that approached the vertices shared by two subtypes are considered to express a hybrid subtype state. Patient samples are color-coded gray if they failed to pass the subtype score thresholds in ≥ 50% of bootstrap iterations. **b** Confusion matrix showing unsupervised clustering results in each classification subtype. **c** Clustering results in Glia-TD and Glia-Ox hybrids. Source data are provided as a Source Data file.

poorly characterized transcripts (Ensembl IDs) – was not significantly linked to any gene ontologies, although a general association with transcription is perhaps a reasonable interpretation. ALS-TD specific expression was observed for many of the features comprising the maroon eigengene (Fig. 3d). Subtype-specificity for eigengene expression was assessed using the β coefficient from dummy regressions considering subtype as the binary predictor and sample-wise eigengene expression as the response (Fig. 3b).

### Patient classification highlights hybrid subtype states
We observed some evidence for the co-expression of subtype phenotypes within this ALS cohort, guided by the clustering, enrichment, and network results (Fig. 1a, f; Fig. 2; Fig. 3a, b). Therefore, to better understand the transcriptional landscape of these molecular subtypes of ALS, we leveraged the classification approach outlined by Patel et al.[23]. Subtype scores were calculated using predictor gene sets derived from the ALS-Glia (gold), ALS-Ox (navy), and ALS-TD (maroon) eigengenes (Fig. 3b; Supplementary Data 7; additional details in Methods section). We observed that the majority of classified patient samples demonstrated gene expression characteristic of a single subtype (220/244; Fig. 4). However, for a subset of patients, hybrid gene expression characteristic of both the ALS-Glia and ALS-TD subtypes ($n = 19$), as well as the ALS-Glia and ALS-Ox subtypes ($n = 5$) was observed. Interestingly, despite shared disease themes between the ALS-Ox and ALS-TD groups (Fig. 2b–d, h), these two subtypes are generally expressed independently. Furthermore, no patient samples were seen to express all three subtypes simultaneously, evident by the fact that all samples fall along one of the three faces of the hexagonal

plot (Fig. 4a). Sample subtypes obtained from the unsupervised clustering analysis are encoded as border colors, and generally show agreement between the two approaches (Fig. 4b). All patient samples shown to express a hybrid ALS phenotype were initially clustered into one of the two subtypes comprising the hybrid state (Fig. 4c), further supporting the interpretation of this analysis. Taken together, the results capture the heterogeneous spectrum of ALS disease phenotypes in this cohort and reveal that a subset of ALS postmortem cortex transcriptomes show evidence for hybrid subtype states.

We further developed four different supervised classifiers[24,25] to assess the ability to stratify new patients, given the postmortem frontal or motor cortex transcriptome (additional details in Methods section). As may be expected given the bootstrap-based classification results (Fig. 4), sensitivity and specificity metrics were relatively poor for all classifiers constructed (Fig. S6).

### The ALS-Glia subtype is associated with a worse prognosis
Next, we considered patient clinical parameters in the context of our subtypes. A survival analysis[26] was performed to determine whether the three molecular subtypes of ALS capture some of the clinical heterogeneity seen in patient disease duration. ALS patients ($n = 208$) were only assigned a subtype if there was a majority consensus among frontal and motor cortex samples or a single tissue sample was characterized for a given patient (additional details in Methods section; Supplementary Data 9). Importantly, we observe that multiple tissue samples from the same donor are classified as the same subtype (80.8%; 126/156), lending support to our subtype assignment methodology.

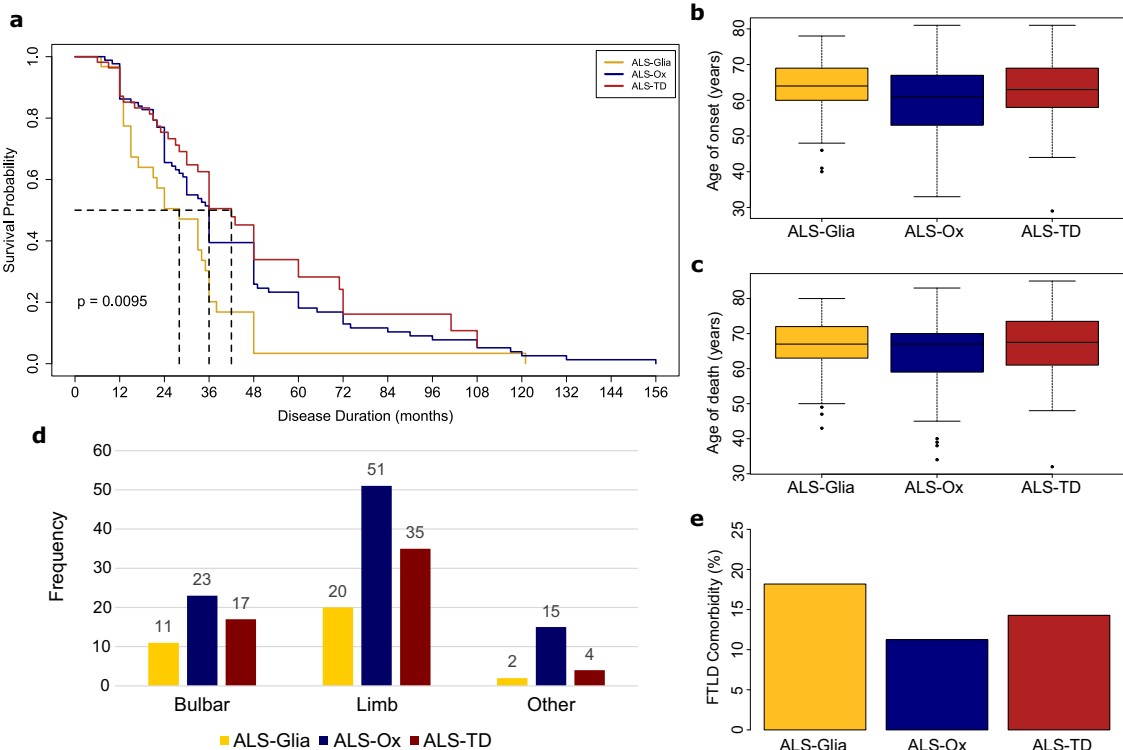

**Fig. 5 | Assessment of ALS patient clinical parameters in the context of disease subtypes. a** Kaplan–Meier survival for the three identified ALS subtypes, with $n = 150$ patients. Patients without an available age of onset or disease duration were excluded from this analysis. The ALS-Glia subtype is significantly associated with a shorter survival duration ($p < 0.01$, log-rank test). The ALS-Ox subtype had a median survival duration of 36 months, while the ALS-TD group had the longest median survival (42 months). **b** Age of disease onset plotted as boxplots for the three ALS subtypes, with $n = 151$ patients. No significant differences are observed in age of onset by subtype. The median is indicated by the solid black line, and first and third quartiles are captured by the bounds of the box. Boxplot whiskers are defined as the first and third quartiles –/+ interquartile range times 1.5, respectively, and outliers are denoted as solid black points. Minimum and maximum values are captured by the lowermost and uppermost points, respectively, or whisker bound if no outliers are shown. **c** Age at death plotted as boxplots for the ALS-Glia, ALS-Ox, and ALS-TD subtypes, with $n = 178$ patients. Again, no significant differences are observed. **d** ALS subtype site of symptom onset, with the 'Other' category comprising axial (4), axial-limb (2), bulbar-limb (4), axial-bulbar (1), generalized (1), and unknown (9) sites of onset. **e** FTLD comorbidity was converted to a percentage and plotted as a bar graph. A Chi-square test of independence was used to assess whether ALS subtype and FTLD comorbidity were associated ($p = 0.59$, one-tailed). Source data are provided as a Source Data file.

Notably, the results show significant differences in patient survival, with the ALS-Glia subtype associated with the shortest disease duration and a median survival of 28 months (Fig. 5a). Pairwise comparisons using the log-rank test showed significant differences in survival between ALS-Glia and ALS-Ox subtypes ($p = 0.015$) and ALS-Glia and ALS-TD subtypes ($p = 0.0043$) but not between the ALS-Ox and ALS-TD subtypes ($p = 0.30$). Consideration of patient age of symptom onset showed a nonsignificant trend toward the latest disease onset for the ALS-Glia subtype ($63.2 \pm 1.83$ years; presented as mean ± standard error) and earliest disease onset for the ALS-Ox subtype ($60.4 \pm 1.16$ years; Fig. 5b, Table S1). We observed the oldest median age at death for the ALS-TD subtype ($66.7 \pm 1.33$ years) and the youngest median age at death for the ALS-Ox subtype ($64.0 \pm 1.05$ years), which likely reflects some dependency on the age of symptom onset (Fig. 5c, Table S1).

Site of symptom onset shows roughly the same proportion of patients with bulbar and limb onset across the three subtypes (Fig. 5d). Subtype comorbidity for FTLD was analyzed using a Chi-square test of independence, although subtype dependency in the co-presentation of ALS and FTLD was not observed ($p = 0.59$). The clinical parameter analysis is further supported by WGCNA results (Fig. 3b), given the ALS-Glia subtype shows the oldest median age of onset and a significantly shorter disease duration – as captured by the gold eigengene (Figs. 5a, b and Table S1). Taken together, these results lend support to the hypothesized existence of subtype-driven clinical heterogeneity in ALS neurodegeneration.

This analysis was also performed with ALS patients that were classified as having a different subtype in each tissue sample transcriptome, termed ALS-Discordant[7] (Fig. S7; Supplementary Data 9). Similar results were observed, with significant differences in patient survival ($p < 0.05$) and the latest age of onset maintained for the ALS-Glia subtype (nonsignificant). We further considered patient clinical parameters in the context of the hybrid subtypes identified in our classification analysis (Fig. S8). In addition, given a large number of patient transcriptomes shared between this cohort and the Tam et al.[7] study, we assessed the agreement of subtype labels for the 140 samples in common (additional details in Methods section; Fig. S1; Supplementary Data 1). We observed 85% agreement (119/140) in sample classification (table S2), despite differences in the features used for patient stratification.

## Subtype-specific gene expression

To provide additional insight into subtype-specific gene expression, a univariate analysis was performed, considering the 1681 genes and TEs used in classification, enrichment, and network construction (additional details in Methods section). Transcript counts were normalized using DESeq2 size factor estimation[14] and $log_2$ transformed (additional details in Methods section). The heatmap and violin plots reflect ALS-Glia (Fig. 6, Fig. S9), ALS-Ox (Fig. 6, Fig. S10), and ALS-TD (Fig. 6, Fig. S11) specific gene and TE expression (Fig. S12; Supplementary Data 10). Out of the 36 transcripts selected to support the characterization of these distinct ALS phenotypes (Fig. 6), 33 were found to have a

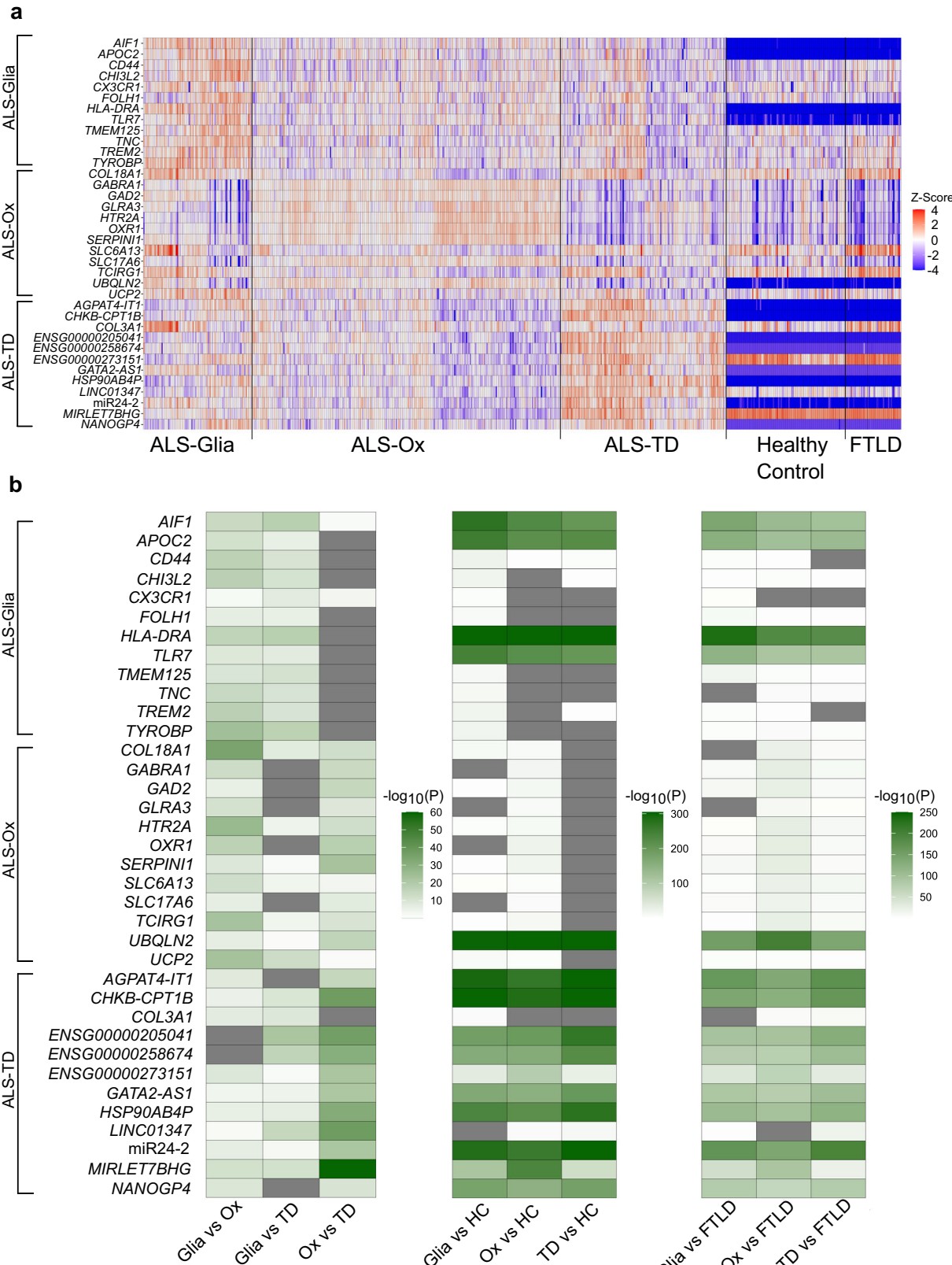

**Fig. 6 | Subtype-specific gene expression. a** Heatmap showing expression of 36 subtype-specific transcripts for all patient samples considered in this study. Count values are adjusted for RIN, site of sample preparation, and sequencing platform covariates. Expression is z-score normalized, using ALS patient expression to define the mean and standard deviation. Control samples with a z-score < −4 are adjusted to −4 for plotting purposes. **b** Presentation of FDR-adjusted *p*-values following pairwise differential expression analysis. *P*-values are −log₁₀ transformed prior to plotting. Gray colored entries indicate an adjusted *p*-value > 0.05. *P*, DESeq2[14] differential expression using the negative binomial distribution, two-tailed, FDR method for multiple hypothesis test correction. Source data are provided as a Source Data file.

distinctive expression in a single subtype, independent of the RNA-seq platform used for analysis (Supplementary Data 10). A few features show rather large differences in normalized expression between control and ALS groups, which may suggest simple thresholding could be used to distinguish the two cohorts (Fig. S13). To support these findings, a univariate analysis was performed, considering FTLD controls and ALS-FTLD patients exclusively. Despite shared pathological mechanisms in these two cohorts, ALS-FTLD patients maintain distinct expression of features presented in Fig. S13 (Fig. S14). Prudencio et al.[10] previously considered the expression of truncated *STMN2* in this cohort, therefore we extended this analysis by considering truncated and normal length *STMN2* (Fig. S15a-d), as well as *C9orf72* and *SOD1* mutation frequency (Fig. S15e), in the context of the identified subtypes. Although no subtype was seen to characteristically express truncated *STMN2*, ALS-Ox samples had significantly upregulated expression of the full length *STMN2* transcript.

Given the significant differences in patient survival (Fig. 5a), subtype-specific gene expression (Fig. 6) may offer a molecular-based approach to inform ALS patient prognosis. Many of these genes and transcripts have not been previously associated with ALS neurodegeneration, offering additional insight into disease pathologies and potential targets for diagnostic or therapeutic development.

**ALS-Glia.** In the ALS-Glia subtype, we note significantly elevated expression of microglia, astrocyte, and oligodendrocyte marker genes (*AIF1*[7], *CCR5*[27], *CD44*[28], *CD68*[29] (Fig. S9), *CHI3L2*[30], *CR1*[31] (Fig. S9), *CX3CR1*[32], *HLA-DRA*[33], *MSR1*[34] (Fig. S9), *TLR7*[35], *TMEM125*[36], *TNC*[36], *TREM2*[18], and *TYROBP*;[18,37] Fig. 6). ALS-Glia upregulation of *CHI3L2*, *CX3CR1*, *FOLH1*, *HLA-DRA*, *ALOX5AP*, *CCR5*, *CR1*, *FPR3*, *NCF2*, *TLR8*, and *TNFRSF25* generally indicates a pro-neuroinflammatory and pro-apoptotic disease phenotype[30,31,38–46] (Fig. 6 and Fig. S9). ALS-Glia negative enrichment for PI3K/AKT signaling further supports a pro-apoptotic disease phenotype[47] (Fig. 3a).

Elevated expression of *TREM2*, *TYROBP*, and *CLEC7A* (Fig. 6, Fig. S9) may suggest a compensatory neuroprotective mechanism, where the activated (DAM) microglia state enhances phagocytic clearance and slows neurodegeneration[18,48]. The DAM phenotype is also known to promote ROS generation and neuroinflammation[49], obscuring the relationship between disease-associated microglia and ALS-Glia pathogenesis. Alterations to lipid metabolism in the ALS-Glia subtype are evidenced by *APOBR*, *APOC1*, and *APOC2* overexpression compared to ALS-Ox and ALS-TD patients (Fig. 6, Fig. S9), and may further reflect the elevated *APOE* and *LPL* expression seen in disease-associated microglia[18,50]. Interestingly, we note upregulated expression of transcripts *CX3CR1*, *TYROBP*, and *TREM2* in this subtype, possibly suggesting dysregulation or competition between homeostatic and activated microglia phenotypes[18] (Fig. 6). Similarly, we observe increased expression disease associated astrocyte[51] (DAA) marker genes in the ALS-Glia subtype, including *ITIH3*, *KCNIP4*, *PDGFD*, *ST6GALNAC5*, and *TNC*. Interestingly, ALS-Glia expression of DAA genes suggests the astrocyte population in these patients captures both disease-associated and homeostatic phenotypes when compared to healthy control donors.

Consistent with the ALS-Glia subtype, we observe characteristic expression of many Fc-gamma receptors and MHC Class II molecules (Fig. 3c, Supplementary Data 10). Heightened *VRK2* expression suggests some anti-apoptotic regulation occurs in ALS-Glia patients[52] (Fig. S9). Overexpression of *FOLH1* may provide evidence for glutamate excitotoxicity susceptibility in the ALS-Glia subtype[38] (Fig. 6). Elevated transcription of *ST6GALNAC2* suggests alterations to post-translational protein O-glycosylation (Fig. S9), while *NINJ2* expression may support the proclivity for neuronal damage and death (Fig. S9). Although additional work is needed to better understand the consequences of the apparently dichotomous microglial phenotypes in the ALS-Glia frontal and motor cortex, these results clearly demonstrate that a subset of ALS patients are defined by glial activation and elevated inflammatory signaling.

**ALS-Ox.** The ALS-Ox subtype is defined by oxidative stress, evidenced by upregulated expression of *OXR1* and *SOD1* and downregulation of *CP* (ceruloplasmin), *UCP2*, and oxidative phosphorylation genes *NDUFA4L2*, *TCIRG1*, and *COX4I2*[53–58] (Fig. 6, Fig. S10). *NDUFA4L2* and *BECN1* expression further implicate impaired autophagy in ALS-Ox pathology[56,59]. We observe subtype-specific expression of many synaptic signaling-associated genes, including *GABRA1* (GABA receptor), *GABRA6*, *GAD2* (catalyzes production of GABA), *GLRA2* (glycine receptor), *GLRA3*, *HTR2A* (serotonin receptor), *KCNV1* (voltage-gated ion channel), *KCNMB1*, *PCSK1*[60], *SLC6A13* (GABA transporter), *SLC17A6* (glutamate transporter), *SLC17A8* (glutamate transporter), and *TCIRG1* (proton transporter associated with synaptic vesicle formation[53]) (Fig. 6, Fig. S10; Supplementary Data 10). Together, the upregulated transcription of *GABRA1*, *GABRA6*, *GAD2*, *GLRA2*, and *GLRA3* and downregulation of *SLC6A13* strongly suggest increased inhibition in the ALS-Ox frontal and motor cortex. Increased expression of *SLC17A6* and *SLC17A8* is hypothesized to reflect a neuronal process to alleviate reduced excitability. Elevated transcription of *BECN1*, *PFDN4*, *SERPINI1* (neuroserpin), *UBQLN1*, and *UBQLN2* suggests proteotoxic stress is also a defining characteristic of this ALS subtype[7,59,61–63] (Fig. 6, Fig. S10; Supplementary Data 10).

Downregulation of *NOS3*, *NOTCH3*, *MYH11*, *MYL9*, and *TAGLN* may implicate pericyte and vascular smooth muscle cell dysfunction and alterations to the blood-brain barrier in ALS-Ox patients[64–66] (Fig. S10). Similar to the ALS-Glia subtype, *B4GALT6* overexpression suggests changes to the O-glycosylated proteome (Fig. S10). Evidence for alterations to the extracellular matrix, in the frontal and motor cortex of ALS-Ox patients, is observed in the downregulated expression of *ADAMTSL4*, *ADAMTS7*, *ADAMTS14*, *COL1A1*, *COL1A2*, *COL2A1*, *COL3A1*, *COL4A6*, *COL6A3*, *COL8A1*, *COL14A1*, *COL18A1*, and *TAGLN* (Fig. 6, Fig. S10; Supplementary Data 10). Interestingly, Collins et al. demonstrate that alterations to the extracellular matrix persist at the protein level[67]. Briefly considering common disease themes between Alzheimer's disease and the ALS-Ox subtype, expression of oxidation-associated transcripts *COX4I2*, *NDUFA4L2*, and *OXR1* is consistent with reported literature[68–70], although *CP* is known to be upregulated in Alzheimer's[55] (Fig. 6). Interestingly, we observe upregulated transcription of *GABRA1*, *GAD2*, *HTR2A*, and *PCSK1* in ALS-Ox patients, which have been previously reported to be downregulated in Alzheimer's patients[71], suggesting distinct synaptic signaling pathological mechanisms (Figs. 3a, 6). Taken together, these results generally suggest ALS-Ox patients reflect more traditional neurodegenerative themes, such as oxidative and proteotoxic stress, impaired blood-brain barrier function, and alterations to synaptic signaling.

**ALS-TD.** The defining characteristic of ALS-TD patients is the dysregulation of transcription, evident by the overexpression of pseudogenes (*EGLN1P1*, *ENSG00000213197*, *HSP90AB4P*, *KRT8P13*, *NANOGP4*, *RPS2OP22*), intronic and antisense transcripts (*AGPAT4-IT1*, *GATA2-AS1*, *TUB-AS1*, *ENSG00000205041*, *ENSG00000263278*, *ENSG00000268670*, and *ENSG00000273151*), long non-coding RNA (*LINC00176*, *LINC00638*, *LINC01347*), and nonsense-mediated decay mRNA (*ARHGAP19-SLIT1*, *C1QTNF3-AMACR*, *CHKB-CPT1B*, and *SLX1B-SULT1A4*) (Fig. 6, Fig. S11; Supplementary Data 10). Upregulated expression of microRNAs miR24-2, miR219A2, miR3648-1, and *MIR-LET7BHG*, relative to the other ALS subtypes, provides additional support for transcriptional and translational dysregulation in ALS-TD patients (Fig. 6, Fig. S11; Supplementary Data 10). miR24-2 has been previously shown to participate in many diseases, including neurodegeneration, serving to regulate cellular proliferation, differentiation, and apoptosis[72]. miR219A2 is known to modulate oligodendrocyte differentiation and remyelination and has been previously reported to

be downregulated in the brains of Alzheimer's patients[73,74]. *MIR-LET7BHG* (*LET-7B* host gene) is also known to regulate gene expression and has been shown to interact with glial receptor *TLR7* to promote neurodegeneration[75]. Therefore, downregulation of *TLR7* in the ALS-TD subtype (Fig. 6) may reflect a neuroprotective state. Altered expression of transcription factors *NKX6-2* and *RUNX3*, relative to controls, further emphasizes transcription as a central pathological mechanism in ALS-TD patients (Fig. S11; Supplementary Data 10).

Similar to the ALS-Ox subtype, we observed the downregulation of transcripts encoding extracellular matrix proteins (Fig. 6, Fig. S11; Supplementary Data 10) and characteristic expression of some transposable elements (Fig. S12). Surprisingly, *TARDBP* (encoding TDP-43) transcription was not a defining feature of ALS-TD patients, and expression was relatively conserved across ALS subtypes, with only moderate upregulation observed compared to healthy controls (Fig. S4b). Transcription of *ADAT3* in ALS-TD patients suggests that the pathological dysregulation of transcription and translation extends to tRNAs[76] (Fig. S11). Consistent with the ALS-TD phenotype, elevated expression of many novel mRNA transcripts was observed, with some examples being *ENSG00000258674*, *ENSG00000279233*, *ENSG00000279712*, *ENSG00000228434*, *ENSG00000234913*, and *ENSG00000250397* (Fig. 6, Fig. S11; Supplementary Data 10). Downregulation of *TP63* suggests alterations to *TP53* signaling and an anti-apoptotic phenotypic state in the ALS-TD subtype[77] (Fig. S11). This interpretation is further supported by the survival analysis (Fig. 5a), given ALS-TD patients demonstrated the longest median disease duration. Taken together, these results suggest poor control of gene transcription and translation in ALS-TD frontal and motor cortices and provide additional insight into the role of TEs in this subtype.

## Cell deconvolution supports upregulated neuroinflammation as a hallmark of the ALS-Glia phenotype

In an effort to address potential biases during bulk tissue sequencing, where varying proportions of glial and neuronal cell types persist, we performed cell deconvolution using CIBERSORTx[78], with DESeq2 normalized count values (additional details in Methods section) and reference single cell expression from Nowakowski et al.[79] (Fig. 7). Ten cell-type signatures (including "Unknown") were generated from the single-cell expression and used to estimate cell percentages in the bulk expression data. Significant differences between prefrontal and motor cortices are observed in microglial, glial progenitor, vascular cell, and inhibitory neuron fractions (Fig. 7a). Weak significant differences are observed in the excitatory neurons, and no significant differences are seen in the astrocytes. Taken together, these findings indicate cell percentages in the frontal and motor cortex may partially explain the subtype-specific expression, although tissue region in the CNS does not strongly influence our assignment of ALS subtype (Fig. S16a). When considering cell type percentages in each subtype, some significant differences were observed (Fig. 7b). The ALS-Ox subtype had a greater average percentage of excitatory neurons as compared to the ALS-Glia subtype (Bonferroni-adjusted *p*-value <1E-5). Similarly, the ALS-Ox subtype demonstrated a significantly greater percentage of inhibitory neurons as compared to the other two subtypes. These results suggest that the ALS-Ox phenotype is partially driven by bulk tissue cell fractions, yet these differences are small in the case of ALS-Ox versus ALS-TD patients, supporting neuronal stress and altered inhibition as hallmarks of the ALS-Ox subtype. The percentage of endothelial and mural cells in ALS-Ox postmortem cortices suggests expression implicating blood-brain barrier dysfunction may be driven by bulk tissue biases. Some significant differences in microglial fraction are observed between the Glia and Ox subtypes (Bonferroni-adjusted *p*-value <1E-9) and Glia and TD subtypes (Bonferroni-adjusted *p*-value <1E-7), suggesting that differences in cell type fractions may, in part, explain the elevated expression of microglial marker genes in ALS-Glia patients. However, it is important to emphasize that no significant

differences in astrocyte fraction were observed between the ALS-Glia subtype and the other two subtypes, indicating that upregulated neuroinflammatory signaling in ALS-Glia patients remains a defining characteristic. Cell deconvolution was also performed on the healthy control and FTLD patients, with results presented in Fig. S16b.

## Discussion

In this study we demonstrate that a large cohort of ALS patient transcriptomes[10] can be stratified into three subtypes defined by distinct molecular phenotypes, termed ALS-Glia[7], ALS-Ox[7], and ALS-TD. Gene expression associated with activated glial cells are observed in the ALS-Glia subtype, while the ALS-Ox subtype is characterized by oxidative stress, proteotoxic stress, and increased inhibition in the frontal and motor cortices. Consideration of locus-specific transposable elements revealed that both the ALS-TD and ALS-Ox subtypes strongly overexpressed TEs compared to healthy control donors and ALS-Glia patients. Guided by enrichment, we observed unique expression of transcription and translation-associated genes, including transcription factors, regulatory microRNAs, mRNA traditionally marked for nonsense-mediated decay, pseudogenes, antisense, intronic, and long non-coding RNAs. These findings led us to define the final subtype by transcriptional dysregulation. These subtypes had significant differences in survival, and the eigengene analysis provides additional insight into the variability observed in ALS patient age at symptom onset and age at death. Given these results, ALS-Glia specific upregulation and downregulation of genes in the frontal and motor cortex provides a set of transcripts associated with patient prognosis.

Noteworthy findings, differing from the foundational Tam et al. study, include (i) our redefinition of the transposable element subtype – driven primarily by our consideration of transposable elements at the gene locus level, (ii) our identification of an immunological eigengene significantly correlated with age of disease onset and survival, and (iii) our observation that the ALS-Glia subtype is associated with a significantly shorter survival duration. However, despite the redefinition of the transposable element subtype, we generally observe good agreement with Tam et al. with respect to the major pathological themes identified in each subtype. Furthermore, given Tam et al. demonstrate TDP-43 binds and regulates a variety of non-protein coding genes, including intronic, long non-coding and regulatory RNA, transposable elements, and intergenic DNA our results suggest that TDP-43 plays a core role in the ALS-TD phenotype.

Importantly, the identified relationship between elevated inflammatory gene expression and shorter disease duration, in ALS-Glia patients, is well supported by previous works[80–82]. Using ALS mouse models expressing mutant *SOD1*, Beers et al.[80] and Biollée et al.[81] both show that microglia become activated and accelerate disease progression, while Yamanaka et al.[82] leveraged Cre-mediated gene excision to demonstrate astrocytes also modulate progression through microglial activation. While these studies do not find associations between glial activation and disease onset, our WGCNA analysis captures a significant positive correlation between inflammatory gene expression and age of onset – potentially a consequence of differences in sample size between these works and our own. Lending additional support to our findings, a recent preprint[83] considering spinal cord samples from the same cohort[10] identified activated microglia modules (eigengenes) negatively correlated with disease duration. Among the inflammatory genes associated with ALS, the chitinases (*CHIT1*, *CHI3L1*) have been considered extensively, with many groups demonstrating that elevated expression is linked to ALS progression and disease duration[84–88]. Consistent with these studies, and others[30], we show that elevated expression of another member of the chitinase family, *CHI3L2*, is uniquely upregulated in ALS-Glia frontal and motor cortices. More generally, activated microglia and astrocytes are known to promote cytotoxicity in motor neurons[89,90], providing a direct framework linking the neuroinflammatory phenotype in ALS-Glia patients to more rapid disease progression.

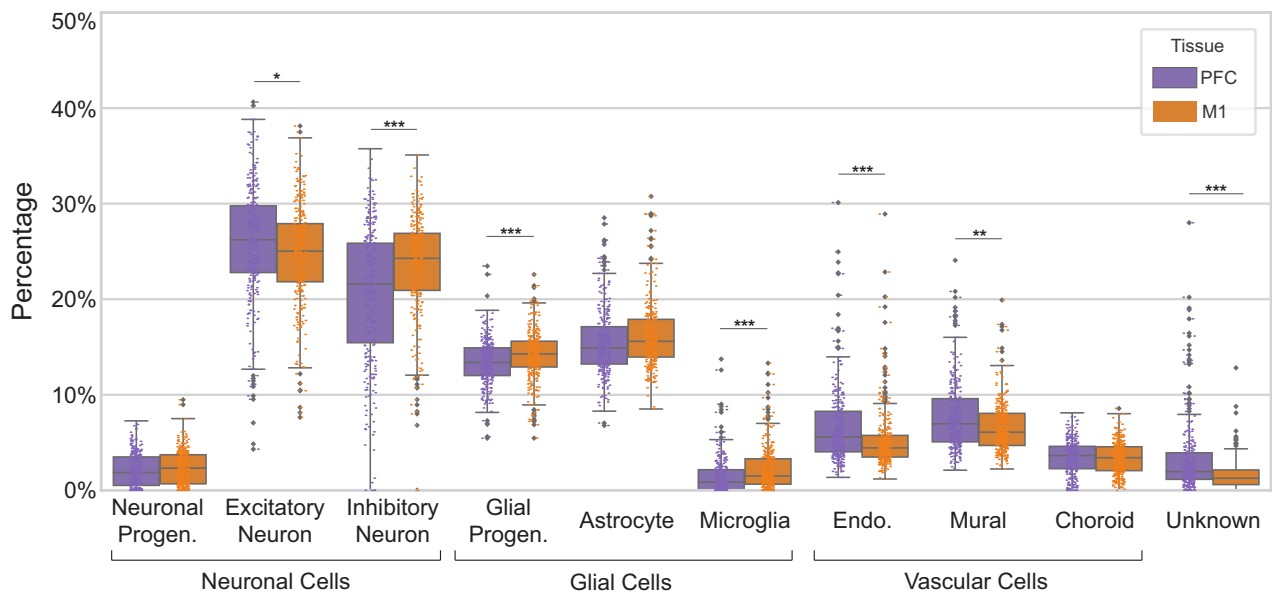

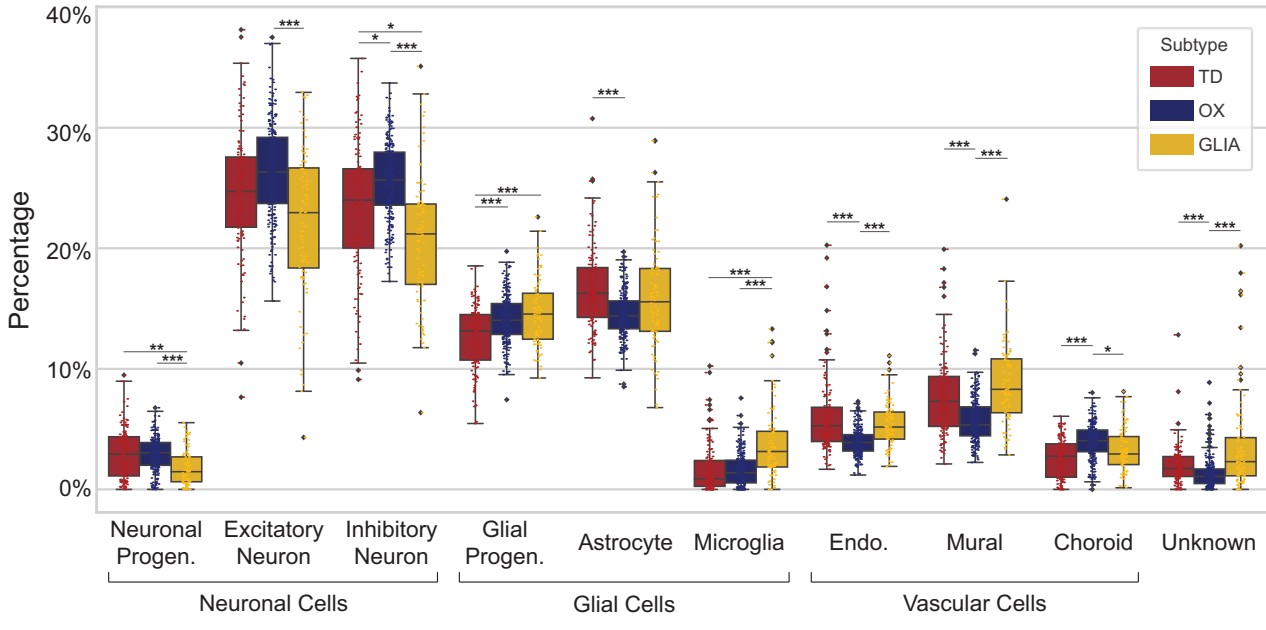

**Fig. 7 | Bulk tissue cell deconvolution in ALS subtypes. a** Cell type percentages in the prefrontal and motor cortices for all patient samples considered in this study. **b** Fractions of cell types in the frontal and motor postmortem cortex, considered in the context of the ALS subtypes. Significant differences in cell type percentages were assessed using a two-sided Wilcoxon rank sum test with Bonferroni *p*-value adjustment. Adjusted *p*-values are denoted using the following scheme: *** $p < 0.001$; ** $p < 0.01$; * $p < 0.05$. The median is indicated by the solid black line, and first and third quartiles are captured by the bounds of the box. Boxplot whiskers are defined as the first and third quartiles ± interquartile range times 1.5, respectively, and outliers are denoted as solid black points. Minimum and maximum values are captured by the lowermost and uppermost points, respectively, or whisker bound if no outliers are shown. Source data are provided as a Source Data file, including exact *p*-values for all comparisons.

Typical neurodegenerative themes dominate the expressed phenotype in ALS-Ox patients. However, we also observe altered expression of synaptic signaling genes, coherently suggesting increased inhibition in the frontal and motor cortex at the end stage of the disease. In contrast, transcranial magnetic stimulation[91] shows ALS

patients present with cortical hyperexcitability early in the pathology, possibly resulting from a combination of increased excitability and decreased inhibition. While these findings appear at odds, a few works considering mRNA expression of inhibitory genes in the frontal and motor postmortem cortex[92,93] find elevated expression of *GAD* and the

β1-subunit of the GABA$_A$ receptor, lending support to the possibility that cortical inhibition shifts from an impaired to overactive state throughout the disease course in response to hyperexcitability. Alterations to brain vascular function, supported by the down-regulated expression of *NOS3, NOTCH3, MYH11, MYL9*, and *TAGLN*, is another distinguishing pathological feature in ALS-Ox patients. This finding is well supported by previous works considering the blood-brain barrier in ALS pathology, with Henkel et al. noting ultrastructural alterations to the blood-brain and blood-spinal cord barriers prior to symptom onset and decreased expression of tight junction proteins in the postmortem sporadic ALS lumbar spinal cord[94]. Similarly, Garbuzova-Davis et al. report pericyte degeneration and endothelial cell damage in medulla and spinal cord tissue from sporadic ALS patients[95], while Saul et al. utilize RNA-seq to reveal changes to the blood-CSF barrier at the level of the choroid plexus[96]. Expanding on these previous findings, our work suggests blood-brain barrier disruptions extend to the frontal and motor cortices late in the disease pathology, implicating systemic vascular changes in the CNS – although cellular deconvolution results suggest ALS-Ox expression could be partially explained by cell type composition.

Neumann et al. demonstrated TDP-43 hyperphosphorylation and mislocalization is a nearly ubiquitous features of ALS and FTLD-TDP pathology[97]. Building on this work, other groups have shown that the TDP-43 protein plays a direct role in the regulation of transcription, including chromatin assembly[98], binding and regulation of transposable elements, intergenic, lncRNA, and intronic DNA[7,99], cryptic exon splicing in *STMN2*[10,100] and *UNC13A*[101,102] genes, and polyadenylation[103]. In line with these findings, our patient stratification analysis identified transcription as the major driver of the ALS-TD phenotype. Elevated expression of nonsense-mediated decay transcripts CHKB-CPT1B and SLX1B-SULT1A4 in ALS-TD patients may implicate a TDP-43 associated mechanism similar to those detailed in the process of cryptic exon splicing in *STMN2* and *UNC13A*[100–103]. Should this be the case, nonsense-mediated decay of read-through genes *CHKB* and *CPT1B* suggests deficits in mitochondrial lipid metabolism, known to play a pathogenic role in muscular dystrophy[104], while loss of *SLX1B* and *SULT1A4* indicate impaired genome stability and monoamine synthesis. Similarly, increased expression of retrotransposons, intronic, antisense, and long non-coding RNA implicates the TDP-43 protein in phenotypic presentation, given findings from Tam et al. and others[7,99]. Of interest, both Brown et al.[101] and Rosa Ma et al.[102] report incomplete detection of the *UNC13A* cryptic exon in ALS patients from the same cohort[10]. Brown et al. observed that 38% of ALS patients expressed the cryptic exon in the *UNC13A* gene, while Rosa Ma et al. found 6.8% (31/454) of frontal and motor cortex samples had detectable levels of the *UNC13A* cryptic exon. In our analysis, we find that 26.9% of patients dominantly expressed the transcriptional dysregulation phenotype, potentially linking ALS-TD patients and cryptic exon expression in *UNC13A* – although additional work is needed to determine if cryptic exon expression is specific to ALS-TD. Despite our observation that *TARDBP* (encoding TDP-43) expression was relatively conserved across subtypes (Fig. S4b), our enrichment, WGCNA, and univariate results, suggest that TDP-43 pathological mechanisms, stemming from mislocalization, drive the expressed phenotype in ALS-TD patients. Given *TARDBP* expression in this cohort, we hypothesize subtype-specific differences in TDP-43 pathology occur at the protein level, and note support for this reasoning in the observation of TDP-43 hyperphosphorylation by Neumann et al.[97].

Clinical and pathological heterogeneity are well-established features of Amyotrophic Lateral Sclerosis. Heterogeneity in clinical presentation is typically characterized by a region of onset, the mixture of upper and lower motor neuron involvement, and rate of progression[105] – although this scheme often fails to accurately predict patient outcomes[3,4]. As a consequence of the poorly understood clinical heterogeneity in ALS, significant research efforts aimed at unraveling the molecular underpinnings in patients[7–9] and animal and cell models[28,38,54,58,59,106] have implicated a number of disease mechanisms shown to contribute to pathological variability. As detailed by Taylor, Brown Jr., and Cleveland[12], these mechanisms include (1) disturbances in protein quality control, including autophagy, proteasome-mediated degradation, and endosome-lysosome mediated degradation, (2) hyperactivated microglia, (3) decreased energy supply from oligodendrocytes following downregulation of MCT1, (4) glutamate excitotoxicity, (5) disturbances in RNA metabolism, and (6) cytoskeletal defects and altered axonal transport. Importantly, the three subtypes identified in this work directly capture the majority of these proposed mechanisms. Supported by our differential expression results, disturbances to protein quality control (proteotoxic stress) is a defining hallmark of ALS-Ox patients, while the hallmark of the ALS-TD patient phenotype is dysregulated RNA metabolism. In ALS-Glia patients, we observe upregulation of inflammatory genes, implicating activated microglia and astrocytes in the accelerated progression of disease pathology. Beyond the major pathological themes of each subtype, we observe some evidence for cytoskeletal defects and altered axonal transport in ALS-Ox and ALS-TD patients through the expression of *ACTA2, DYNLT3, PLS1*, and *TUBB6*. Moderate overexpression of *FOLH1* implicates glutamate excitotoxicity in ALS-Glia patients, although further consideration of transcripts and proteins associated with glutamate metabolism and signaling are needed to explore subtype specificity. In summary, this work helps to clarify the molecular foundation of clinical and pathological heterogeneity in ALS by demonstrating that subtype-specific phenotypes are associated with patient outcomes, including survival and age of onset.

## Methods

### Study approval
The NYGC ALS Consortium samples presented in this work were acquired through various IRB protocols from member sites and the Target ALS postmortem tissue core and transferred to the NYGC in accordance with all applicable foreign, domestic, federal, state, and local laws and regulations for processing, sequencing, and analyses[10].

Postmortem brain tissues from patients with FTLD-TDP or PSP and from cognitively normal individuals were obtained from the Mayo Clinic Florida Brain Bank. Diagnosis was independently ascertained by trained neurologists and neuropathologists upon neurological and pathological examinations, respectively. Written informed consent was given by all participants or authorized family members, and all protocols were approved by the IRB and ethics committee of the Mayo Clinic[10].

### Data sources
**ALS patients.** Within the GEO data repository, GSE153960 was identified as the ideal study to further probe the existence of ALS subtypes. GSE153960 contains RNA-seq data from 1659 tissue samples, spanning 11 regions of the CNS, from 439 patients with ALS, frontotemporal lobar degeneration (FTLD), or comorbidities for ALS-Alzheimer's (ALS/AD) or ALS-FTLD. These 1659 tissue samples were filtered such that only the individuals belonging to the groups ALS-TDP, ALS/FTLD, ALS/AD, and ALS-SOD1 were considered. Furthermore, RNA-seq samples derived from regions of the CNS other than the frontal or motor cortex, such as cerebellum and spinal cord, were not included in the analysis - yielding 473 cortex transcriptomes (Fig. S1a).

Raw FASTQ files for the 473 ALS patient samples were downloaded from the European Bioinformatics Institute data repository (NCBI mirror) using NIH's Globus software. Of the 473 selected RNA-seq samples, five had incomplete or missing paired-end FASTQ files, and were subsequently excluded from the analysis. An additional 13 samples were mapped to the human reference genome build hg38 via STAR 2.5.3a[107] but TEs were not successfully quantified using the SQuIRE pipeline and were, therefore, excluded from the analysis. A final 4 samples were poorly mapped to the RepeatMasker transposable

element reference genome[108,109]; retaining these four subjects would have resulted in a reduction of 'shared' TEs by > 60% (557/1474). Our final ALS cohort contained 451 frontal and motor cortex transcriptomes, corresponding to 208 unique patients ($n = 95$ female, $n = 113$ male). Subject demographics for this analysis are included in table S1. A full list of the included and excluded samples is provided in Supplementary Data 1.

**Control subjects.** Control sample transcriptomes were comprised of healthy control donors (HC; $n = 93$) and patients diagnosed with FTLD exclusively ($n = 42$), corresponding to 58 HC and 42 FTLD individuals. Equivalent to the ALS subject processing pipeline, raw FASTQ files were downloaded from the European Bioinformatics Institute data repository. One RNA-seq sample had missing paired-end FASTQ files and was excluded from our analysis. The remaining 135 control samples were mapped to the human reference genome build hg38 using STAR 2.5.3a and TEs were quantified using SQuIRE's Count function. TEs missing from our control sample cohort were replaced with a count value of 0.

Transcriptomes from the control cohort were implemented during GSEA for the identification of enriched pathways associated with each of the three subtypes. Control samples were further utilized to assess differentially expressed genes and TEs in each ALS subtype. Control transcriptomes were subject to cell deconvolution in an effort to assess bulk tissue RNA-sequencing biases.

**Quantification.** Quantification of gene expression was performed using RSEM[110], as detailed by Prudencio et al.[10]. The processed gene count matrix was accessed directly from the GEO Accession (GSE153960) and counts were rounded to integers as recommended by the authors of RSEM and required by DESeq2 differential expression.

SQuIRE[13] (Supplementary Data 3) was selected for transposable element quantification, as this alignment pipeline provides locus-specific TE counts, allowing for a deeper analysis beyond TE subfamilies. Similar to RSEM, SQuIRE applies the Expectation Maximization (EM) algorithm to optimize the allocation of multi-mapped reads. SQuIRE's Fetch, Clean, Map, and Count functions were utilized to align and quantify locus-specific transposable elements. The EM 'tot_counts' values were selected as the estimate for sequencing reads attributed to the transposable elements. The hg38 build was used during mapping, with default trim and EM parameters, and a read length of 100 or 125 base pairs depending on the sequencing platform specified. A scoring threshold of $\geq 99$ was used to restrict the number of false positive TEs (1%), with few uniquely mapping reads. Only the locus-specific TEs with at least one count for all ALS samples ($n = 451$) were included in downstream analysis, resulting in 1474 unique TE features (Supplementary Data 2). The naming scheme for our locus-specific transposable elements is presented in SQuIRE[13], however in brief, TE feature names included the mapping chromosome, start and stop base pairs, transposable element subfamily, family and superfamily identifiers, base mismatches in parts per thousand, and sense or antisense strand annotation.

**Differential expression.** As discussed by Prudencio et al., the large ALS cohort size required the utilization of two different sequencing platforms (HiSeq 2500 and NovaSeq 6000, Illumina, San Diego, CA) to complete the analysis. Exploratory differential expression considering sequencing platforms as the design equation factor revealed strong batch effects in gene expression, evident by more than one-third of all genes falling below the Benjamini-Hochberg corrected p-value threshold (37.2%, 22478/60403; including TEs). To correct for these batch effects, we followed the approach outlined by Prudencio et al. and split our ALS cohort based on the sequencing platform. Our NovaSeq cohort contained 255 patient transcriptomes ($n = 106$ female, $n = 149$ male), while our HiSeq

cohort contained 196 ($n = 97$ female, $n = 99$ male). The control cohort was processed in an analogous manner.

DESeq2[14] (Supplementary Data 3) was initially applied to perform a preliminary differential expression on gene and TE counts. Differential expression was utilized to guide the removal of sex-dependent genes prior to clustering. As described by Prudencio et al., sex was determined using XIST and UTY expression. Default parameters were used for DESeq2 differential expression, with male specified as the reference level and the 'betaPrior' argument in the DESeq() function set to true. A Benjamini-Hochberg corrected $p$-value $\leq 0.05$ was selected as the threshold for the removal of sex-dependent genes.

**Clustering.** Following the removal of sex-dependent genes using the differential expression, the raw count matrix was subject to a variance stabilizing transformation (VST) to address heteroskedasticity in gene counts[14]. The VST counts were then subject to rank ordering by median absolute deviation (MAD) and the top 10,000 features were retained for unsupervised clustering analysis by non-negative matrix factorization (NMF)[15,111]. This process was completed for both sequencing platform cohorts.

**Rank estimation.** Factorization rank was estimated in R, Version 4.0.3 (The R Foundation for Statistical Computing, Vienna, Austria) using the NMF package[112] (Supplementary Data 3). We selected a rank of three for clustering analysis, based on the plots of the cophenetic correlation coefficient for ranks spanning 2 to 6. Quality measures were estimated using 50 iterations at each rank and the default seeding method. The nsNMF (non-smooth non-negative matrix factorization) method was utilized for all NMF clustering[15].

**Non-negative matrix factorization.** Non-negative matrix factorization was performed in SAKE, a convenient tool for RNA-seq sample pre-processing, filtering, clustering and visualization[113] (Version 0.4.0). The top 10,000 MAD genes, after a variance stabilizing transformation, were utilized as the input into SAKE. No samples were removed during the quality control step, and further transformations in the filtering step were not necessary. During non-negative matrix factorization, selected parameters include factorization rank = 3, iterations = 200, and NMF method set to nsNMF.

To robustly assign ALS sample subtypes, 10 rounds of NMF clustering were performed in SAKE. For each patient sample, the ALS subtype with a simple majority was selected. For a small number of edge cases (5/451), an eleventh round of NMF clustering was used as a tiebreaker to reach the simple majority threshold. This process was completed for both sequencing platform groups. The robustly assigned subtype labels are provided for all ALS patient samples (Supplementary Data 11).

**Feature selection.** After each replicate of NMF clustering, gene and TE feature scores[114] were calculated for all 10,000 MAD transcripts. Feature scores were averaged across the 10 clustering replicates and reordered. The top 1000 features from both sequencing platform cohorts were combined, and after the removal of duplicates, 1681 genes and TEs remained for enrichment, networking, and univariate analysis.

**Enrichment analysis.** Following supervised classification, the gene and TE feature sets were then enriched using Enrichr[16] and GSEA[17] (Version 4.1.0, Broad Institute, Boston, MA). Enrichr was performed to support subtype-specific pathway expression observed during GSEA, utilizing the Fisher's exact test with Benjamini–Hochberg multiple hypothesis test correction. Hypergeometric enrichment analysis was considered in the context of the Reactome 2016 database. Upregulation and downregulation of pathways was determined using subtype-specific differential expression, with each feature assigned to two of

the three subtypes based on the maximum and minimum median expression. For GSEA, healthy control donors were selected as the reference phenotype during enrichment. Transcripts without a corresponding gene symbol (HGNC) were excluded from the enrichment analysis, including TEs, leaving 891 total genes. The minimum gene set size was adjusted to 5, and all other parameters were maintained as the default. For the enrichment, we leveraged the canonical pathways contained in the Reactome database[115], a custom gene set containing markers of disease-associated microglia[18,116], and curated gene sets for Alzheimer's, Parkinson's, and ALS[53,117]. Pathway heatmaps reflecting gene enrichment by phenotype were built using the Rank Metric Score tabulated during GSEA.

A custom gene set for the enrichment of locus-specific transposable elements was also considered, however, GSEA rank-based scoring may be biased by the size of the TE set (>400 features). The collapse of locus-specific TEs to the subfamily level was also considered, to allow enrichment using Repbase[109], however, subfamily co-expression was not observed following a hierarchical clustering analysis considering TE features exclusively (Supplementary Data 6).

**Networks.** Network development was carried out in two different, yet complementary approaches. For the visualization of gene enrichment pathways by ALS phenotype, we leveraged Cytoscape (Version 3.8.2, Institute for Systems Biology, Seattle, WA)[19]. Result files from GSEA were utilized as the input into Cytoscape. Additional pathway enrichment was performed using the custom and curated gene sets from the previous step. Nodes were color-coded according to ALS subtype specificity, guided by GSEA enrichment score magnitude and univariate analysis. A small number of unrelated or synonymous pathways were manually trimmed.

Co-expressed gene sets associated with disease duration, age of symptom onset, and age at death were assessed using the Weighted Gene Co-Expression Network Analysis (WGCNA) package in R[20] (Version 1.70-3, University of California, Los Angeles). The minimum module size was set to 25 and a soft power of 13 was selected given the assessment of scale-free topology. All 1681 features were considered during the construction of the eigengene heatmap, using variance-stabilizing transformation count values. Eigengenes were assessed for upregulation or downregulation in each subtype using dummy regression, with subtype as the predictor and sample-wise eigengene expression as the response variable. For each eigengene, a linear regression model was constructed, setting one of the three subtypes to a value of 1, and the other two to a value of 0. The sign and magnitude of the β coefficient from the non-zero term reflect subtype-specific eigengene expression.

Eigengenes of interest were subject to network visualization in VisANT[22] using edge weights derived from WGCNA (Supplementary Data 12). A weight threshold of 0.05 was set to filter genes weakly co-expressed. Unconnected nodes were manually trimmed from the networks.

**Classification.** Given that previously established predictor gene sets for ALS subtype were not available, ALS-Glia, ALS-Ox, and ALS-TD predictor gene sets were derived from our gold, navy, and maroon eigengenes, respectively (Fig. 3b). We utilized enrichment results from WGCNA and differential expression to establish subtype-specific expression of each eigengene. For example, most transcripts comprising the gold eigengene are specifically upregulated in the ALS-Glia subtype (Fig. 3c; Fig. S5c). Transcript counts were considered on the DESeq2 median-of-ratios scale, adjusted for RIN, site of collection, and sequencing platform covariates.

Subtype scores, defined as the average expression of subtype-specific predictor genes minus the average expression of all 1681 features considered in this analysis, were calculated for 100 different sets of predictors (per subtype) and used to define a 5% cutoff for the

expected subtype score[23]. Each sampled predictor gene set contained the same number of features as the original eigengene and were generated by randomly sampling the eigengenes with replacement. For example, the expected subtype score for ALS-Glia patients was determined by first generating 100 predictor sets by randomly sampling features comprising the gold eigengene. Then, the average (sample-wise) ALS-Glia expression was determined for each of the 100 predictor sets and subtracted from the average (sample-wise) ALS-Glia expression of all 1681 classification genes.

After repeating this analysis for the ALS-Ox and ALS-TD subtypes, using their respective eigengenes, 100 subtype scores were generated for all 451 samples ($n = 203$ female, $n = 248$ male). A 5% cutoff for the expected subtype score was then established, per sample, and final subtype classification thresholds were determined by weighting expected subtype scores according to the observed proportion of patient samples in each subtype (obtained from clustering). Bootstrapping was then applied, involving the sampling of predictor gene sets (with replacement) and the calculation of subtype scores for 1000 iterations.

Patient samples were initially placed at the origin, and moved in the direction of the subtype vertex after passing the corresponding subtype threshold. Therefore, the x, y, and z-axis vertices reflect the expression of a single subtype, while the other three vertices capture a combination of two subtypes. Individual points that passed a given subtype threshold in >50% of bootstrap iterations were filled with their respective subtype colors. Samples were considered to express a hybrid subtype state if one subtype threshold was passed >50% of the time and simultaneously passed a second subtype threshold >40% of the time. One ALS-TD sample (CGND-HRA-01732) did not have a RIN value available and was subsequently excluded from the analysis due to an incomplete design equation.

All machine learning classifiers were developed in Python (Version 3.8.8, Python Software Foundation, Wilmington, DE) using the Scikit-learn framework[24] (Version 0.24.1). Four different models were considered, k-nearest neighbors (KNN), linear support vector classification (Linear SVC), multilayer perceptron (MLP), and random forest (RF). To limit the inclusion of platform-dependent genes, the top 1000 features were further filtered so that only genes and TEs shared between the two sequencing platform cohorts were retained, totaling 299. The k-nearest neighbor classifier was built with k neighbors = 5, distance calculated using the Manhattan metric, weights = distance, and all other parameters as default. The linear SVC classifier was constructed using class weights defined by the proportion of subtypes in the NovaSeq cohort, max iterations = 100,000 and default for all other parameters. The multilayer perceptron neural network was built using three hidden layers (five total), with 100 'neurons' comprising each hidden layer, learning rate = 0.0001, hyperbolic tangent activation function, random state = 1, max iterations = 10,000 and default settings for all remaining parameters. Finally, the random forest was developed using n estimators = 1000, oob score = True, class weights defined by the proportion of subtypes in the NovaSeq cohort, and default for all other parameters. All models were constructed using the 'one-vs-rest' multi-class strategy.

Supervised classifiers were constructed using training and testing datasets generated from a 70% / 30% split of the ALS NovaSeq cohort ($n = 255$ transcriptomes). 100-fold cross validation was applied to assess performance in the testing cohort. The ALS HiSeq cohort ($n = 196$ transcriptomes) was designated as the holdout dataset to assess performance metrics when classifying new patient samples. Transcript counts on the VST scale were utilized during classifier development. Classifier recall, precision, and F1 scores were calculated for all ALS subtypes after each round of cross validation.

**Clinical parameters.** For many patients in our cohort, multiple tissue samples from the frontal and motor cortex were characterized by RNA-seq (Supplementary Data 9). As a result,

patients were assigned a label only if there was a majority consensus among their frontal and motor cortex samples, or if there was a single sample characterized. ALS patients which displayed multiple subtypes among their frontal and motor cortex samples were labeled 'Discordant'. Among the 208 unique patients in this cohort ($n = 95$ female, $n = 113$ male), 30 were found to be discordant (table S1, S10; $n = 17$ female, $n = 13$ male). Differences in ALS survival by subtype were assessed using the Kaplan-Meier analysis[26,118] with application of the log-rank statistical test. Subtype-specific differences in age of symptom onset and age at death were analyzed using ANOVA tests. A Chi-squared test of independence was applied to assess subtype specificity for FTLD comorbidity. All analysis was performed with and without discordant ALS patients.

**Subtype concordance.** Both the Prudencio et al.[10] and Tam et al.[7] studies are associated with the New York Genome Center (NYGC) ALS Consortium (GEO Superseries GSE137810), so a large majority (~95%; $n = 140$; $n = 77$ female, $n = 63$ male) of postmortem tissue samples analyzed by Tam et al. are also reanalyzed by Prudencio et al. We took advantage of this repeat analysis by utilizing the work from Tam et al. as a reference to assess patient subtype concordance.

**Univariate analysis.** Transcript counts were normalized using DESeq2[14] size factor estimation (median-of-ratios) to better allow comparison between patient samples. Subtype-specific differential expression of transcripts was determined using a multifactor design equation, accounting for sequencing platform, RIN, and site of sample collection covariates. One ALS-TD sample (CGND-HRA-01732) did not have a RIN value available and was subsequently excluded from the analysis due to an incomplete design equation. Pairwise analysis was performed using the constrast() argument, for all combinations. Genes and TEs with an FDR adjusted $p$-value $\leq 0.05$ were considered to be significant. All patient samples ($n = 586$; $n = 267$ female, $n = 319$ male) were considered during normalization. Counts on the median-of-ratios scale were $\log_2$ transformed before plotting. For heatmap presentation, z-scores were calculated using ALS patients to establish gene-wise mean expression and deviation, with expression values on the $\log_2$ median-of-ratios scale. FDR adjusted $p$-values, derived from DESeq2 differential expression were $-\log_{10}$ transformed prior to plotting.

A few additional genes not included in the 1681 features used for classification, enrichment, and networking, were also considered during the univariate analysis out of disease relevance[7,97] and include *TARDBP, OXR1, BECN1, BECN2, SOD1, UBQLN1, UBQLN2, UCP2,* and *TXN*. Many of these added genes were used during unsupervised clustering as some of the top 10,000 most variable features calculated by median absolute deviation.

**Cell deconvolution.** Cell deconvolution was performed using CIBERSORTx[78] with reference single cell RNA-sequencing expression from the developing human brain available from Nowakowski et al.[79] (http://bit.ly/cortexSingleCell). Raw data were filtered and normalized. 35 cell types (WGCNAcluster) were grouped into 10 major cell types: neuronal progenitor, excitatory neuron, inhibitory neuron, glial progenitor, astrocyte, microglia, endothelial, mural, choroid, and unknown. Marker genes for each major cell type were identified using Seurat's[119] function FindAllMarkers() (Version 4.0.3). Marker genes were used to generate medioids (i.e., cell type signatures) to use as the reference for cell deconvolution. The ALS cohort was normalized using DESeq2 with count values on the median-of-ratios scale. All overlapping MAD transcripts between the NovaSeq and HiSeq cohorts were used, totaling 7372 transcripts, to ensure a sufficient number of transcripts were available for deconvolution. Transcripts without a mapped gene symbol and transposable elements were removed from the analysis which led to 4912 transcripts. Lastly, transcripts not shared between ALS and control

cohorts ($n = 586$; $n = 267$ female, $n = 319$ male) and Nowakowski cell type signatures were removed. 1881 transcripts remained and were used as input into CIBERSORTx. Quantile normalization was disabled in CIBERSORTx, which is recommended for RNA-seq data, and 500 permutations were used for significance analysis. Significant differences in cell type fractions were assessed using the nonparametric Wilcoxon rank sum test with Bonferroni correction.

### Reporting summary
Further information on research design is available in the Nature Portfolio Reporting Summary linked to this article.

## Data availability
The raw data files used in this study are available in the NCBI Run Selector database under accession code PRJNA644618. The RSEM processed gene count matrix utilized in this study are available in the Gene Expression Omnibus database under accession code GSE153960. Processed RNA-seq count files utilized during our analysis are available as supplemental tables or made publicly available at: https://figshare.com/authors/Jarrett_Eshima/13813720. Source data are provided with this paper.

## Code availability
All code developed and utilized in this analysis is available in the Barbara Smith Lab Github repository[120] (https://github.com/BSmithLab/ALSPatientStratification), excluding some scripts used for supervised classification, which can be found in the Plaisier Lab Github repository (https://github.com/plaisier-lab/U5_hNSC_Neural_G0).

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

## Acknowledgements

The authors would like to acknowledge The Target ALS Human Postmortem Tissue Core, New York Genome Center for Genomics of Neurogenerative Disease, Amyotrophic Lateral Sclerosis Association, Tow Foundation, and the patients and family members for supporting this analysis. The authors would like to acknowledge Solo Pyon for his assistance with the high-speed download of patient sequencing data from the European Bioinformatics Institute data repository (NCBI mirror). The authors thank Paula Phan for her assistance with vector-based figure preparation. The authors would like to thank Oliver. H Tam and Molly G. Hammell for their correspondence regarding unsupervised clustering. The authors would also like to thank Yu-Jui Ho for his assistance with SAKE usage and installation. All NYGC ALS Consortium activities are supported by the ALS Association (ALSA, 19-SI-459) and the Tow Foundation. J.E. is supported by the National Science Foundation, Graduate Research Fellowship (026257-001).

## Author contributions

J.E. and B.S.S. proposed the research question addressed in this study. J.E. and S.A.O. developed all code utilized to analyze data. R.B., C.L.P., and B.S.S. supervised the project and guided experimental methodology. J.E. performed the formal analysis and E.M., S.A.O., and J.E. generated all figures. J.E. and B.S.S. wrote the manuscript, and all authors provided edits prior to submission.

## Competing interests

R.B. is the chairman of the board of Iron Horse Diagnostics, which has not contributed financially, or by any other means, to this study. The remaining authors declare no competing interests.

## Additional information

## NYGC ALS Consortium

Robert Bowser ⓘ [2,3]

A full list of members and their affiliations appears in the Supplementary Information.

