## [Peer Review File · Nature Communications]

Molecular subtypes of ALS are associated with differences in patient prognosisREVIEWER COMMENTS

Reviewer #1 (Remarks to the Author):

Eshima and colleagues re-analyse a large dataset of transcriptomes from the NYGC ALS Consortium. Combining estimates of gene and transposable element expression from frontal and motor cortex, they use non-negative matrix factorisation to split the cohort into three subclusters. Although this is an approach performed previously by Tam et al (on a smaller subset of samples), Eshima and colleagues are able to extend these findings. By using the available clinical metadata on the cohort, they identify subtype associations with age of onset and disease duration, finding that the ALS-Glia subtype, characterised by increased expression of inflammatory and glial marker genes, have a shorter disease duration, a very interesting result.

Altogether I find it an engaging and clearly written piece of work, replicating and extending a previous study with some novel findings. I have a few suggestions for how the authors could strengthen the work.

Major comments

In general, the figures are far too small. Minimum font size needs to be increased to allow readability. Any text used in a figure should be readable, otherwise it should be removed. Figures 3 and 6 are particularly egregious for this.

There are several non-disease factors that may partially explain the subtype classifications that I would like the authors to address:

- Submitting site/hospital (due to differences in sample preparation and storage)**
- RNA integrity number and other metrics of RNA quality - 3' or 5' bias, % exon/intron coverage**
- Brain region (frontal cortex, medial vs lateral motor cortex)**

Did the authors apply a cut-off for lowly expressed genes? My concern is that some of the apparent subtype differences are driven by noisy gene expression. Can the authors rule this out?

It is reassuring that multiple tissue samples from the same donor are more often than not classified as the same subtype. I would suggest that the authors more strongly highlight this reproducibility between repeated samples of the same donor.

It is a common analysis to apply a deconvolution algorithm to estimate the proportions of the major cell-types within a bulk RNA-seq sample. In human cortex samples one can estimate the proportions of the major cell-types and compare proportions across groups of samples. I am curious whether the cell-type proportions differ between the three subgroups, given your findings of altered microglia and astrocyte marker genes in the ALS-Glia subtype.

Increased expression of inflammatory genes has previously been observed to associate with shorter disease duration, particularly the CHIT1 gene, as well a recent preprint using the spinal cord samples from this same cohort. I would suggest that you include these previous observations in your discussion of this result. Have the authors considered including the spinal cord samples from the NYGC cohort as a way of validating the subtyping in the cortex?

The increased sample size relative to Tam et al has allowed the authors to examine multiple clinical variables in association with their ALS subtype labels. I'm curious why the authors did not also include associations with the presence of C9orf72 repeat expansions or SOD1 mutations, the two most common genetic causes of ALS.

Furthermore, the expression of the TDP-43 regulated cryptic splicing event in STMN2

has already been quantified in these samples (Prudencio et al, 2020). Given the transcriptional dysregulation subtype, I am curious why the authors did not look at the presence of truncated STMN2 splicing in their subtypes? Nor do they discuss the ongoing series of works directly linking TDP-43 mislocalisation to specific sites of transcriptional regulation, including cryptic splicing, polyadenylation and TE binding.

In Figure 3, the networks of pathways, edges denote sharing of genes. What does sharing with TEs mean? Is the TE network all a set of TEs rather than genes?

This reviewer does not find dendrograms (3C) and network visualisations (3D-E) helpful. I would strongly suggest the authors rethink how they present their WGCNA network results.

Reviewer #2 (Remarks to the Author):

In the manuscript entitled "Molecular subtypes of ALS are associated with differences in patient prognosis", Eshima et al. analyzed a large set of RNA-seq data from the frontal and motor cortices of ALS patients to distinguish molecular subtypes of ALS based on common gene expression signatures, which also included transposable elements (TEs) mapping uniquely to loci and quantified. They defined three ALS subtypes: ALS-Glia, ALS-Ox, and ALS-TD. Using Gene Set Enrichment Analysis to contrast each subtype's expression profile against control cortices, they described some unique and some shared pathways among the subtypes. From this data set, they also constructed co-expression networks and highlighted modules comprised to subtype-specific genes. They developed a score-based classification approach based on these gene sets to classify ALS subtypes, and they refined the molecular subtypes of ALS present in their data set, demonstrated by observing that some samples were classified as hybrid subtypes. Finally, the authors assessed whether subtype classification through gene expression clustering could distinguish clinical features of the ALS patients. They demonstrated that the ALS-Glia subtype is significantly associated with a shorter survival duration.

Overall, this work applies several sophisticated methods to analyze transcriptomics data towards revealing insights into the heterogeneous nature of ALS. Like many consortia-based studies, this work aims to identify reliable biomarkers that can accurately diagnose and inform clinical decisions on ALS management and treatment. The noteworthy features of this manuscript are the large number of patient samples incorporated into the analyses and the management and reporting of the meta data. However, there are several concerns that limit the impact of the work overall.

This study seems to be an extension of the foundational work by Tam et al., 2019. <https://pubmed.ncbi.nlm.nih.gov/31665631/>, as the authors consistently reference. While many findings were reproduced, such as the classification of the 3 ALS subtypes and the gene expression pathways enriched in each, there are also different conclusions drawn. These include the observation that TARDBP is not reduced in the ALS-TD subtype (wherein Tam et al. the reduced expression of TARDBP explains the loss of repression of transposons) and the shorter survival of ALS-Glia subtypes (where Tam et al. observe no differences). How can the authors reconcile these different findings?

While the data presented mostly supports the conclusions made by the authors, analyses could strengthen the support for their conclusions. Also, there are some flaws in the application of some analyses, as well as unclear descriptions of the methods (Details about this are explained in the specific comments on the figures and sections below).

Methods

Clustering: It is not clear if all or some of the 1475 TEs are included in the 10,000 most

variable features used for clustering. Also, what is the extent of overlap of the 10,000 features between the HiSeq and Novaseq data sets?

Feature Selection: Why were the top 1,000 features selected from the 10,000 MAD transcripts? The remaining 1681 genes and TEs used for enrichment and network building are much lower than what is typically used for these analyses. <https://www.ncbi.nlm.nih.gov/pmc/articles/PMC2756411/> GSEA was performed on only 891 genes. For gene sets that are up to 500 genes (the default limit), which is over half of the 891 genes, this can likely lead to false enrichment. How was the ranked list generated?

Classification: This section is not clear. Perhaps a description of an example predictor selection would help here. The R scripts on the Github repository are accessible, however the Rdata and csv input files are not.

Overall, the Figures are presented in low resolution and text was difficult to read.

Figure 1: Do ALS/FTLD cases segregate?

Figure 2: A-C: Are these enriched gene sets with adjusted p-values at a > 0.05 threshold? Is this a comprehensive list of significant gene sets? **D:** to what extent do these TEs overlap with those analyzed by Tam et al.? Are these the same TEs bound by TDP43?

Figure 3: Break down each module by the percentage of TEs present in each. Could that explain why Magenta module does not enrich for gene ontologies?

Figure 4: The value of this analysis is low. NMF was already used to define the 3 clusters. From this, the modules were selected based on their associations with each subtype, and the classifiers were taken from the modules, and then reapplied to classify samples from the same data set. This is self-referential. Patel et al. <https://pubmed.ncbi.nlm.nih.gov/24925914/>, defined classifier genes from TGCA and applied these to classify inn new data sets of single cells in tumors. Despite this re-classification, what more can be said about the hybrid states phenotypically? Do they have altered survival, age of onset, etc? Visually, it is difficult to distinguish border colors from fill colors on the plot. Consider showing histograms and quantification of the classifications.

Figure 5: A: the outcome of this analysis is different from Tam et al. This is worth a mention and analysis in the Discussion section.

Figure 6: FTLD comorbidity ALS cases should be analyzed separately if they are to be compared to FTLD cases.

Figure S5: B: Would the results be the same for the 142 overlapping samples in Tam et al.? How do you reconcile the different findings? Were the same TEs quantified between the two studies? Is it due to platform or VST differences?

Figure S6: B-E: Consider performing Wilcox Rank sum to assess significance of prediction accuracy.

Figure S7: A: Is it fair to include survival curves for subjects with only 1 tissue profiled? These patients are classified with lower confidence than those patients with frontal and motor cortices both classified as the same subtype.

Figure S12: If these are strong candidates as markers for distinguishing FTLD and ALS/FTLD, consider supporting this idea by testing them as predictors in a validation data set.

Reviewer #3 (Remarks to the Author):

Esthima et al. address the question of heterogeneity in ALS by reanalyzing bulk RNAseq data from the motor/frontal cortex of over 200 patients and performing a patient stratification analysis. They identified 3 molecular subtypes, ALS- Glia (glia activation); ALS-Ox (oxidative stress and altered synaptic signaling), and ALS-TD (transcriptional dysregulation), and linked them to the age of onset and survival as clinical outcomes. Although interesting, there are concerns regarding the methodology used to identify subtypes and their clinical relevance:

1) The question addressed is very complex, however the paper does not consider some important factors, both clinical/biological metadata as well as potential technical artifacts, that can affect RNAseq data. For instance:

- Region: The authors classify the regions used as "frontal" and "motor" cortices; what region(s) of the frontal lobe were used? Different frontal regions have different cellular compositions and are not equally affected in ALS (or in cohort of FTLN patients used as controls).

- Although data from different brain regions are available from this cohort, the authors used only frontal/motor. Including "control" regions that are not affected in ALS such as the cerebellum is helpful to address covariates such as the RIN, the postmortem interval, agonal stage, and genetic factors

- Clinical and pathological heterogeneity is described in ALS, how do the three subtypes identified by this paper relate to those?

2) Cellular composition is usually a driver of clustering in the analysis of bulk RNAseq, and so clusters unbiasedly identified may reflect the relative abundance of specific cell types. Cell type deconvolution methods to infer cell type proportions may help address this. This is important in this paper since two of the subtypes of ALS identified correspond with glial (ALS-Glia) and neuronal (ALS-Ox) responses.

3) Differential gene expression needs to take into account potential covariates beyond sex.

Response to Reviewers' Comments:

The authors would like to thank the reviewers for their consideration of our work and interest in our findings. The authors have implemented reviewers' suggestions for improvement, and the following changes have been made to the manuscript text and supplemental file:

Figures:

- Fig. 2 – Enrichment analysis is reconsidered using the Fisher's exact test. Enriched pathways with an adjusted p-value < 0.05 are now shown in Fig. 2A. Figure has been edited to improve text size.
- Fig. 3 – WGCNA results have been presented in a different way to enhance informational value. The original Fig. 3C-E have been moved to the supplemental as fig. S6. Subtype-specific eigengenes (purple, magenta, and turquoise) have been redefined for added clarity.
- Fig. 4 – Additional figures (4B-C) have been provided to summarize agreement between clustering and classification methods. Analysis has been updated to account for RIN and site covariates.
- Fig. 5 – Text has been edited to improve readability
- Fig. 6 – Analysis was re-performed, adjusting for RIN, site of collection, and sequencing platform covariates. Violin plots have been replaced by two heatmaps detailing i) sample-wise normalized expression and ii) FDR-adjusted p -values on the $-\log_{10}$ scale to improve visibility.
- Fig. 7 – New figure considering cell deconvolution of bulk tissue RNA-seq in the ALS cohort, using CIBERSORT and reference single cell expression from Nowakowski et al.

Supplementary Figures:

- Original Fig. S4 – Figure S4D and S4F have been moved to manuscript Figure 3 and the remaining plots have been removed, in an effort to focus on the most relevant subtype-associated pathways. Figure S4I from the original supplemental figure has been moved to Fig. S4C.
- Fig. S4A – edited to improve readability.
- Fig. S5 – supplemental figure showing eigengene correlation heatmap and network representations
- Fig. S6 – edited to improve readability
- Fig. S7 – edited to improve readability
- Fig. S8 – new supplemental figure considering clinical parameters in the hybrid subtypes identified in our bootstrap-based classification analysis
- Fig. S9 – edited to improve readability
- Fig. S10 – edited to improve readability
- Fig. S11 – edited to improve readability
- Fig. S12 – edited to improve readability
- Fig. S13 – edited to improve readability
- Fig. S14 – new supplemental figure considering univariate features distinguishing ALS from FTLN patients, in a cohort restricted to ALS-FTLN patients and controls exclusively.
- Fig. S15 – new supplemental figure considering truncated and normal length *STMN2* and known mutations in the identified ALS subtypes.
- Fig. S16 – new supplemental figure reporting cell type fractions in ALS patients and controls.

Supplementary Tables:

- Table S2 – the authors noticed the Dec. 2016 ensembl archive used throughout this analysis is currently not available. To facilitate replication, the authors have provided a condensed look-up-table containing ensembl gene IDs and the corresponding gene symbol. RIN values have been added to sample metadata.
- Table S12 – Differential expression results have been updated to reflect the added consideration of RIN and site of collection covariates.
- Table S13 – Updated phenotype file containing RIN and site of collection covariates

Code and Data Availability:

- All code has been updated in the Github repository, where appropriate. Data files (.csv and .RData) have been made available at: https://figshare.com/authors/Jarrett_Eshima/13813720

Reviewer 1:

Eshima and colleagues re-analyse a large dataset of transcriptomes from the NYGC ALS Consortium. Combining estimates of gene and transposable element expression from frontal and motor cortex, they use non-negative matrix factorisation to split the cohort into three subclusters. Although this is an approach performed previously by Tam et al (on a smaller subset of samples), Eshima and colleagues are able to extend these findings. By using the available clinical metadata on the cohort, they identify subtype associations with age of onset and disease duration, finding that the ALS-Glia subtype, characterised by increased expression of inflammatory and glial marker genes, have a shorter disease duration, a very interesting result.

Altogether I find it an engaging and clearly written piece of work, replicating and extending a previous study with some novel findings. I have a few suggestions for how the authors could strengthen the work.

Major comments

In general, the figures are far too small. Minimum font size needs to be increased to allow readability. Any text used in a figure should be readable, otherwise it should be removed. Figures 3 and 6 are particularly egregious for this.

The authors thank the reviewer for their recommendation. All figures have been edited to improve readability and updated accordingly in the manuscript.

There are several non-disease factors that may partially explain the subtype classifications that I would like the authors to address:

- Submitting site/hospital (due to differences in sample preparation and storage)
- RNA integrity number and other metrics of RNA quality - 3' or 5' bias, % exon/intron coverage
- Brain region (frontal cortex, medial vs lateral motor cortex)

Submitting site/hospital (due to differences in sample preparation and storage)

Site of sample preparation is presented below as a bar chart. Given that the majority of ALS-Ox patients were found to originate from the Target ALS cohort, the authors performed a supplementary differential expression analysis, removing site as a covariate, for the 36 features shown in manuscript Fig. 6 (Table 1).

Figure 1. Bar charts showing ALS subtype site of sample preparation.

Original Design: sequencing platform + subtype

Updated Design: sequencing platform + Site + subtype

Gene	Original: Glia vs Ox	Updated: Glia vs Ox	Original: Glia vs TD	Updated: Glia vs TD	Original: Ox vs TD	Updated: Ox vs TD
AIF1	6.95E-15	4.28E-13	2.58E-23	1.02E-22	2.92E-04	6.95E-15
APOC2	1.71E-10	2.90E-11	6.38E-06	2.84E-06	1.89E-01	1.71E-10
CD44	4.55E-17	2.12E-17	1.51E-10	4.51E-11	2.10E-01	4.55E-17
CHI3L2	1.02E-17	9.84E-18	9.34E-12	5.57E-12	3.51E-01	1.02E-17
CX3CR1	8.26E-04	9.23E-03	1.00E-10	7.04E-10	2.15E-05	8.26E-04
FOLH1	3.25E-06	5.04E-07	7.16E-05	1.70E-05	8.92E-01	3.25E-06
HLA-DRA	2.86E-16	4.50E-16	1.77E-22	5.05E-23	3.84E-03	2.86E-16
TLR7	2.60E-08	1.31E-08	1.19E-07	4.39E-08	7.92E-01	2.60E-08
TMEM125	7.49E-10	8.47E-10	3.59E-10	1.68E-10	4.45E-01	7.49E-10
TNC	3.49E-16	5.03E-15	1.24E-09	1.32E-09	1.64E-01	3.49E-16

TREM2	1.71E-18	2.55E-17	3.56E-09	3.07E-09	1.99E-02	1.71E-18
TYROBP	1.46E-24	2.74E-23	1.98E-17	1.15E-17	4.79E-01	1.46E-24
COL18A1	4.63E-44	6.00E-33	3.17E-04	3.55E-03	2.59E-30	4.63E-44
GABRA1	1.01E-10	1.81E-11	6.80E-02	1.27E-01	1.00E-24	1.01E-10
GAD2	3.84E-03	2.96E-06	7.63E-05	7.36E-04	6.95E-19	3.84E-03
GLRA3	1.12E-11	4.20E-11	9.51E-01	1.00E+00	2.28E-16	1.12E-11
HTR2A	8.31E-24	1.92E-23	1.99E-01	1.38E-01	9.55E-23	8.31E-24
OXR1	1.82E-15	1.53E-14	6.28E-02	8.85E-02	2.38E-33	1.82E-15
SERPINI1	7.05E-06	1.37E-07	3.30E-07	3.49E-06	4.74E-34	7.05E-06
SLC6A13	9.40E-29	1.58E-11	1.90E-05	1.09E-01	5.77E-13	9.40E-29
SLC17A6	5.91E-09	1.18E-07	3.98E-01	3.87E-01	2.59E-16	5.91E-09
TCIRG1	1.40E-28	2.63E-22	5.54E-02	9.26E-02	3.16E-24	1.40E-28
UBQLN2	1.86E-04	1.10E-06	9.25E-04	6.72E-03	1.98E-19	1.86E-04
UCP2	8.33E-21	4.18E-23	1.55E-10	6.47E-12	1.85E-02	8.33E-21
AGPAT4-IT1	1.56E-07	2.02E-08	1.23E-04	3.29E-04	4.01E-30	1.56E-07
CHKB-CPT1B	1.89E-07	2.41E-06	2.54E-15	5.82E-15	3.07E-61	1.89E-07
COL3A1	3.13E-18	9.81E-07	2.19E-13	3.78E-07	6.44E-01	3.13E-18
ENSG00000205041	3.82E-01	3.24E-01	2.69E-30	3.59E-29	2.81E-59	3.82E-01
ENSG00000258674	1.64E-01	2.46E-01	8.05E-25	2.93E-24	3.76E-53	1.64E-01
ENSG00000273151	2.06E-09	1.89E-09	4.45E-06	1.18E-05	9.21E-40	2.06E-09
GATA2-AS1	8.94E-07	5.41E-04	3.03E-08	1.21E-09	2.69E-39	8.94E-07
HSP90AB4P	3.24E-04	1.24E-05	1.14E-16	3.20E-15	4.90E-52	3.24E-04
LINC01347	5.08E-03	7.93E-03	5.41E-21	2.05E-20	1.03E-55	5.08E-03
MIR24-2	1.40E-10	6.79E-07	9.28E-05	1.45E-05	1.20E-37	1.40E-10
MIRLET7BHG	9.33E-15	4.73E-12	4.19E-19	4.19E-19	4.39E-94	9.33E-15
NANOGP4	1.63E-07	1.91E-08	2.56E-02	4.73E-02	1.34E-20	1.63E-07

Table 1. FDR-adjusted p -values for the 36 features included in Figure 6, comparing the original design equation and one including site of collection as a covariate.

Our results show the site of sample collection/preparation does account for some of the variation in subtype expression. However, it did not significantly impact the interpretation of the subtypes. Because the site of collection/preparation did have an impact, we decided it prudent to adjust normalized count values for the site as a covariate and update our differential expression results in the manuscript accordingly (Fig. 6, figs. S10-15, Table S12).

RNA integrity number (RIN) and other metrics of RNA quality – 3' or 5' bias, %exon/intron coverage

RIN values have been added to Table S2 and S13 for all samples considered in this analysis, and summary metrics are provided below for convenience:

Disease Group	Average RIN
ALS Spectrum	6.14
FTLD	5.46
Healthy Control Donor	5.80

ALS Subtype	Average RIN
ALS-Glia	6.35
ALS-Ox	6.47
ALS-TD	5.38

Given ALS-Ox patients primarily originate from the Target ALS cohort and ALS-TD patients show a lower average RIN, we performed an additional differential expression analysis, adjusting for both RIN (centered and scaled) and site covariates. We present FDR-adjusted p -values for the 36 features shown in manuscript Figure 6 (Table 2).

Original Design: sequencing platform + subtype

Final Design: sequencing platform + RIN + site + subtype

Gene	Original: Glia vs Ox	Final: Glia vs Ox	Original: Glia vs TD	Final: Glia vs TD	Original: Ox vs TD	Final: Ox vs TD
AIF1	6.95E-15	3.08E-14	2.58E-23	8.20E-19	2.92E-04	3.72E-02
APOC2	1.71E-10	6.91E-12	6.38E-06	5.08E-06	1.89E-01	6.92E-02
CD44	4.55E-17	1.36E-17	1.51E-10	2.69E-10	2.10E-01	1.16E-01
CHI3L2	1.02E-17	5.96E-18	9.34E-12	3.29E-11	3.51E-01	2.21E-01
CX3CR1	8.26E-04	9.52E-03	1.00E-10	1.84E-07	2.15E-05	1.16E-03
FOLH1	3.25E-06	5.01E-07	7.16E-05	3.84E-06	8.92E-01	1.00E+00
HLA-DRA	2.86E-16	5.58E-17	1.77E-22	8.85E-19	3.84E-03	2.40E-01
TLR7	2.60E-08	5.05E-09	1.19E-07	2.39E-07	7.92E-01	9.40E-01
TMEM125	7.49E-10	6.41E-10	3.59E-10	1.93E-11	4.45E-01	2.88E-01
TNC	3.49E-16	1.08E-14	1.24E-09	1.31E-10	1.64E-01	5.90E-01
TREM2	1.71E-18	7.11E-18	3.56E-09	1.32E-10	1.99E-02	1.27E-01
TYROBP	1.46E-24	1.59E-24	1.98E-17	8.90E-18	4.79E-01	4.69E-01
COL18A1	4.63E-44	3.49E-35	3.17E-04	2.97E-08	2.59E-30	1.37E-12
GABRA1	1.01E-10	1.91E-13	6.80E-02	6.53E-01	1.00E-24	3.54E-14
GAD2	3.84E-03	4.08E-06	7.63E-05	5.50E-02	6.95E-19	1.64E-15
GLRA3	1.12E-11	1.18E-11	9.51E-01	2.32E-01	2.28E-16	3.27E-09
HTR2A	8.31E-24	1.50E-26	1.99E-01	9.59E-05	9.55E-23	1.30E-12
OXR1	1.82E-15	2.32E-17	6.28E-02	6.35E-01	2.38E-33	1.20E-18
SERPINI1	7.05E-06	2.16E-09	3.30E-07	2.69E-02	4.74E-34	7.36E-23
SLC6A13	9.40E-29	1.33E-12	1.90E-05	2.34E-04	5.77E-13	1.46E-03
SLC17A6	5.91E-09	4.95E-07	3.98E-01	9.29E-01	2.59E-16	4.96E-08
TCIRG1	1.40E-28	4.17E-23	5.54E-02	1.57E-04	3.16E-24	2.64E-10
UBQLN2	1.86E-04	1.23E-06	9.25E-04	4.74E-02	1.98E-19	1.22E-16
UCP2	8.33E-21	1.90E-23	1.55E-10	1.73E-13	1.85E-02	4.97E-02
AGPAT4-IT1	1.56E-07	1.93E-08	1.23E-04	2.92E-01	4.01E-30	1.20E-15
CHKB-CPT1B	1.89E-07	1.05E-05	2.54E-15	5.51E-10	3.07E-61	9.84E-38
COL3A1	3.13E-18	1.50E-07	2.19E-13	2.20E-09	6.44E-01	2.04E-01
ENSG00000205041	3.82E-01	7.10E-01	2.69E-30	2.61E-22	2.81E-59	3.25E-37
ENSG00000258674	1.64E-01	4.32E-01	8.05E-25	5.69E-17	3.76E-53	2.86E-31
ENSG00000273151	2.06E-09	2.05E-09	4.45E-06	3.82E-02	9.21E-40	3.23E-22
GATA2-AS1	8.94E-07	1.27E-04	3.03E-08	6.50E-05	2.69E-39	1.31E-21
HSP90AB4P	3.24E-04	2.22E-06	1.14E-16	4.03E-07	4.90E-52	1.77E-32
LINC01347	5.08E-03	1.72E-02	5.41E-21	1.05E-15	1.03E-55	4.74E-38
MIR24-2	1.40E-10	4.06E-07	9.28E-05	3.62E-03	1.20E-37	5.60E-22
MIRLET7BHG	9.33E-15	5.45E-12	4.19E-19	6.75E-12	4.39E-94	4.05E-60
NANOGP4	1.63E-07	9.25E-10	2.56E-02	7.42E-01	1.34E-20	2.69E-10

Table 2. FDR-adjusted p -values comparing the original design equation and one accounting for both site and RIN as covariates. No major differences in differential expression significance were observed for the 36 subtype-specific features.

While the average RIN values are lower than usual for *in vitro* experimentation, previous work shows our values are typical for postmortem brain tissue¹. Next, we used RIN and site as covariates for differential expression analysis. These results show that the subtype specific expression of these features is still observed when the RIN and site are used as covariates. We

did observe that patient site of sample preparation and RIN do capture some of the variability between ALS-Ox and ALS-TD patients.

Therefore, we have elected to use median-of-ratios count values adjusted for RIN, site, and platform covariates and present these updated FDR p -values in Fig. 6, figs. S9-S14 from the manuscript. Table S12 has been updated with FDR-adjusted p -values accounting for RIN and site covariates for all pairwise comparisons. Additional method details have been added to the manuscript in lines 37-39 on page 24. One ALS-TD sample (CGND-HRA-01732) did not have a RIN value provided and was subsequently excluded from the analysis due to an incomplete design equation.

3' bias and percent exon coverage metrics were determined using SAMtools² and Picard (v2.27.4-0) and presented in Figure 2 of this response. Our results show 3' bias is within the range typically reported in previous works^{3,4}, for all patient groups.

Given that transcript count values served as the foundation of this analysis, the percentage of mRNA bases covered during sequencing was determined not to influence the reported counts in the case of uniquely mapping reads. Transcript counts were normalized using DESeq2 size factor estimation, further addressing bias and variance concerns associated with mRNA coverage in weakly expressed genes.

Figure 2. Quality metrics assessed using SAMtools² and Picard. (A) 3' bias, calculated as the mean coverage of the 3' most 100 bases divided by the mean coverage of the whole transcript. A summary statistic is provided for each sample and calculated as the median 3' bias for the top 1000 most highly expressed transcripts. (B) percent mRNA base coverage, calculated as the sum of bases mapping to UTR and coding regions of mRNA transcripts, presented as boxplots.

Brain region (frontal cortex, medial vs lateral motor cortex)

Figure 3. Patient brain region parsed by subtype

We observe approximately the same ratio of Glia, Ox and TD patients in the frontal and specified motor cortices, roughly matching the ratio observed during unsupervised clustering, indicating brain region is not a confounding factor with subtype. Weak dependencies in subtype classification are observed for patient samples derived from an unspecified region of the motor cortex – potentially reflecting variability in tissue collection procedures as compared to the specified motor regions. Figure 3 from this response has been added as a new supplementary figure (Figure S16A) and referenced in the manuscript in lines 16-26 on page 15.

Did the authors apply a cut-off for lowly expressed genes? My concern is that some of the apparent subtype differences are driven by noisy gene expression. Can the authors rule this out?

The authors elected to initially consider some low expressed genes during unsupervised clustering in an effort to consider all of the transposable elements passing our filter threshold (at least one count in all patient samples) as potential disease markers⁵.

The authors further considered expression of the 36 subtype-specific features presented in Fig. 6 in the manuscript, to rule out the possibility that subtype differences are driven by noise. When considering the normalized expression of the 1681 features used for enrichment, WGCNA, and classification, over 85% of the transcripts were observed to have mean expression > 10 normalized counts – a commonly applied threshold to remove lowly expressed genes (Figure 4). All 36 features are observed to have >10 raw counts in at least half of all patient samples (n=451) and the results are shown in Figure 4 and 5 of this response.

Figure 4. Mean of normalized counts adjusted for RIN, site of collection and sequencing platform covariates, presented on the \log_{10} scale, showing the 1681 genes used in enrichment, WGCNA, and differential expression (“Enrichment Features”). All features presented in manuscript Fig. 6 (“Subtype-specific Features”) show mean expression above 10 normalized counts (red dashed line).

Figure 5. Boxplots showing log₁₀ counts for the 36 features presented in Fig. 6 in the text. The red dashed line denotes 10 raw counts. The large majority of features are observed to have median expression above 10 counts.

It is reassuring that multiple tissue samples from the same donor are more often than not classified as the same subtype. I would suggest that the authors more strongly highlight this reproducibility between repeated samples of the same donor.

The authors thank the reviewer for their recommendation. The following sentence has been added to the manuscript in lines 25-27 on page 9:

“Importantly, we observe that multiple tissue samples from the same donor are classified as the same subtype (80.8%; 126/156), lending support to our subtype assignment methodology.”

It is a common analysis to apply a deconvolution algorithm to estimate the proportions of the major cell-types within a bulk RNA-seq sample. In human cortex samples one can estimate the proportions of the major cell-types and compare proportions across groups of samples. I am curious whether the cell-type proportions differ between the three subgroups, given your findings of altered microglia and astrocyte marker genes in the ALS-Glia subtype.

The authors thank the reviewer for the recommendation to consider the effects of cell type proportion in our analysis of bulk tissue RNA-seq. The authors have performed cell deconvolution using CIBERSORT⁷⁸, with DESeq2 normalized count values and reference single cell expression from Nowakowski et al.⁷⁹. In an effort to ensure a sufficient number of features were available for deconvolution, the authors used all overlapping MAD transcripts between the NovaSeq and HiSeq cohorts, totaling 7372 transcripts. Removal of transcripts without a corresponding gene symbol and transposable elements led to 4912 transcripts, 1881 of which were shared between the ALS cohort and Nowakowski single cell reference expression.

Figure 6. Bulk tissue RNA-seq cell deconvolution in ALS subtypes. Fractions of cell types in the frontal and motor postmortem cortex, considered in the context of the ALS subtypes. Significant differences in cell type percentages were assessed using the Wilcoxon rank sum test with Bonferroni p -value adjustment. Adjusted p -values are denoted using the following scheme: *** $p < 0.001$; ** $p < 0.01$; * $p < 0.05$.

Generally, we observe some significant differences in cell type percentages between the three subtypes. In the excitatory and inhibitory neuronal populations, the ALS-Ox subtype had significantly greater percentage of neurons, as compared to the ALS-Glia subtype. Weakly significant differences in inhibitory cell percent are seen between ALS-TD and ALS-Ox patients, yet these differences are not significant in the excitatory neuron population estimate. These findings suggest bulk tissue biases capture some of ALS-Ox specific expression, although it is clear that varying cell type percentages do not fully explain characteristic expression differentiating Ox and TD subtypes. Similarly, cell deconvolution demonstrates ALS-Glia patient samples had a greater estimated fraction of microglial cells and glial progenitor cells, as compared to the other two subtypes. Despite the greater percentage of microglial and glial progenitor cells

in ALS-Glia samples, the ALS-Glia subtype did not demonstrate significantly different percentages of astrocytes, maintaining upregulated neuroinflammation as a defining characteristic.

The manuscript has been updated to include our cell deconvolution analysis, with the results presented on page 15 and 16. The methods section has been updated on page 25 – to describe our approach to this analysis.

Increased expression of inflammatory genes has previously been observed to associate with shorter disease duration, particularly the CHIT1 gene, as well a recent preprint using the spinal cord samples from this same cohort. I would suggest that you include these previous observations in your discussion of this result. Have the authors considered including the spinal cord samples from the NYGC cohort as a way of validating the subtyping in the cortex?

The authors thank the reviewer for their recommendation to more strongly highlight past works associating shorter disease duration with increased expression of inflammatory genes. The following sentences have been added to the discussion section of the manuscript to address this point, lines 1-20 on page 17.

“Importantly, the identified relationship between elevated inflammatory gene expression and shorter disease duration, in ALS-Glia patients, is well supported by previous works^{80,81,82}. Using ALS mouse models expressing mutant *SOD1*, Beers et al.⁸⁰ and Biollée et al.⁸¹ both show that microglia become activated and accelerate disease progression, while Yamanaka et al.⁸² leveraged Cre-mediated gene excision to demonstrate astrocytes also modulate progression through microglial activation. While these studies do not find associations between glial activation and disease onset, our WGCNA analysis captures a significant positive correlation between inflammatory gene expression and age of onset – potentially a consequence of differences in sample size between these works and our own. Lending additional support to our findings, a recent preprint⁸³ considering spinal cord samples from the same cohort¹⁰ identified activated microglia modules (eigengenes) negatively correlated with disease duration. Among the inflammatory genes associated with ALS, the chitinases (*CHIT1*, *CHI3L1*) have been considered extensively, with many groups demonstrating that elevated expression is linked to ALS progression and disease duration^{84,85,86,87,88}. Consistent with these studies, and others³⁰, we show that elevated expression of another member of the chitinase family, *CHI3L2*, is uniquely upregulated in ALS-Glia frontal and motor cortices. More generally, activated microglia and astrocytes are known promote cytotoxicity in motor neurons^{89,90}, providing a direct framework linking the neuroinflammatory phenotype in ALS-Glia patients to more rapid disease progression.”

80. Beers, D.R. et al. Wild-type microglia extend survival in PU. 1 knockout mice with familial amyotrophic lateral sclerosis. *Proceedings of the National Academy of Sciences* **103**, 16021-16026 (2006).

81. Boillée, S. et al. Onset and progression in inherited ALS determined by motor neurons and microglia. *Science* **312**, 1389-1392 (2006).

82. Yamanaka, K. et al. Astrocytes as determinants of disease progression in inherited amyotrophic lateral sclerosis. *Nature Neuroscience* **11**, 251-253 (2008).

83. Humphrey J. et al. Integrative genetic analysis of the amyotrophic lateral sclerosis spinal cord implicates glial activation and suggests new risk genes. Preprint at <https://www.medrxiv.org/content/10.1101/2021.08.31.21262682v1> (2021).

84. Thompson, A.G. et al. CSF chitinase proteins in amyotrophic lateral sclerosis. *Journal of Neurology, Neurosurgery & Psychiatry* **90**, 1215-1220 (2019).

85. Steinacker, P. et al. Chitotriosidase (CHIT1) is increased in microglia and macrophages in spinal cord of amyotrophic lateral sclerosis and cerebrospinal fluid levels correlate with disease severity and progression. *Journal of Neurology, Neurosurgery & Psychiatry* **89**, 239-247 (2018).

86. Illán-Gala, I. et al. CSF sAPP β , YKL-40, and NfL along the ALS-FTD spectrum. *Neurology* **91**, e1619-e1628 (2018).

87. Gille, B. et al. Inflammatory markers in cerebrospinal fluid: independent prognostic biomarkers in amyotrophic lateral sclerosis?. *Journal of Neurology, Neurosurgery & Psychiatry* **90**, 1338-1346 (2019).

88. Vu, L. et al. Cross-sectional and longitudinal measures of chitinase proteins in amyotrophic lateral sclerosis and expression of CHI3L1 in activated astrocytes. *Journal of Neurology, Neurosurgery & Psychiatry* **91**, 350-358 (2020).

89. Liddelow, S.A. et al. Neurotoxic reactive astrocytes are induced by activated microglia. *Nature* **541**, 481-487 (2017).

90. Zhao, W. et al. Activated microglia initiate motor neuron injury by a nitric oxide and glutamate-mediated mechanism. *Journal of Neuropathology & Experimental Neurology* **63**, 964-977 (2004).

The authors elected to restrict analysis to the frontal and motor cortex regions of the CNS given the frontal and motor cortices have been well studied regarding ALS pathology, however further studies should consider additional regions – including the spinal cord. Additionally, supporting evidence for pathological onset in the brain⁶⁻⁸ was hypothesized to better reflect phenotypic variability at the end stage of ALS, although the authors recognize this is still a matter of debate and evidence exists to the contrary⁹.

The increased sample size relative to Tam et al has allowed the authors to examine multiple clinical variables in association with their ALS subtype labels. I'm curious why the authors did not also include associations with the presence of C9orf72 repeat expansions or SOD1 mutations, the two most common genetic causes of ALS.

The authors thank the reviewer for their recommendation to consider common genetic mutations in the context of the ALS subtypes. Given the large number of patients with an “unknown” genetic mutation, the authors had not considered this analysis. Figure 7 in this response presents genetic causes of ALS, separated by ALS subtype. Chi-squared tests of independence show no association between ALS subtype and C9orf72 or SOD1 mutations, although the large number of unknown patient mutations limits the interpretability of this analysis. Figure 7 from this response has been included as a new supplemental figure (S15E) and referenced in the manuscript in lines 20-23 on page 11.

Figure 7. Stacked bar chart showing C9orf72 and SOD1 mutation frequency in the ALS cohort. A chi-squared test of independence was performed to assess mutation dependency on subtype. After removal of the “unknown” categorical variable, the null hypothesis (no association between ALS subtype and common genetic drivers) was accepted for both C9orf72 ($p = 0.47$) and SOD1 ($p = 0.21$). It is important to note that the limited number of observations for SOD1 may drive inaccurate estimation of the chi-squared test statistic.

Furthermore, the expression of the TDP-43 regulated cryptic splicing event in STMN2 has already been quantified in these samples (Prudencio et al, 2020). Given the transcriptional dysregulation subtype, I am curious why the authors did not look at the presence of truncated STMN2 splicing in their subtypes? Nor do they discuss the ongoing series of works directly linking TDP-43 mislocalisation to specific sites of transcriptional regulation, including cryptic splicing, polyadenylation and TE binding.

As discussed by Prudencio et al., truncated STMN2 levels were not associated with TDP-43 'subtype', indicating that tSTMN2 levels are elevated in all types of TDP-43 pathology (FTLD-TDP and ALS-TDP). In addition, truncated STMN2 was not found to have an association with patient survival. Taken together, the findings from Prudencio et al. suggest tSTMN2 would be a poor differentiating marker for ALS subtype and prognosis.

The authors considered truncated STMN2 in the context of our subtypes, although given that a large number of patient samples had 0 tSTMN2 counts reported (Figure 8) we elected to exclude the analysis. However, given the reviewer's recommendation, we have provided this analysis in Figure 9. In addition, we further consider STMN2 expression on the TPM and DESeq2 median-of-ratios scales, and observe significantly elevated expression in the ALS-Ox subtype (Figure 9). It is important to note that Prudencio et al. did not find an association between full-length STMN2 and TDP-43 burden or clinical outcomes, limiting predictive value. Figure 9 from this response has been included as a new supplemental figure (S15A-D) and referenced in the manuscript in lines 20-25 on page 11.

Figure 8. Histograms showing tSTMN2 expression are provided on the (a) TPM and (b) raw count scales.

Figure 9. Consideration of truncated and normal length Stathmin-2 in the context of ALS subtypes. The Mann-Whitney U test was used to assess statistical significance in tSTMN2 expression on both the (a) TPM scale and (b) raw count scale. After adjusting p-values for multiple hypothesis testing, truncated STMN2 expression was not observed to have any association with ALS subtype. (c) Full length transcript STMN2 counts on TPM scale, evaluated using the Mann-Whitney U test, with Bonferroni-adjusted p-values shown. (d) Full length transcript *STMN2* counts on the DESeq2 median-of-ratios scale, evaluated using differential expression, with FDR-adjusted *p*-values shown. Healthy control donors and FTLD patients are included, in an effort to improve the estimation of size factors for normalization.

The authors thank the reviewer for their recommendation to highlight the ongoing works linking TDP-43 mislocalization to sites of transcriptional regulation. The following sentences have been added to the discussion section of the manuscript to address this point, page 17 line 45 – page 18 line 28.

“Neumann et al. demonstrated TDP-43 hyperphosphorylation and mislocalization is a nearly ubiquitous feature of ALS and FTLD-TDP pathology⁹⁷. Building on this work, other groups have shown that the TDP-43 protein plays a direct role in the regulation of transcription, including chromatin assembly⁹⁸, binding and regulation of transposable elements, intergenic, lncRNA, and intronic DNA^{7,99}, cryptic exon splicing in *STMN2*^{10,100} and *UNC13A*^{101,102} genes, and polyadenylation¹⁰³. In line with these findings, our patient stratification analysis identified transcription as the major driver of the ALS-TD phenotype. Elevated expression of nonsense-mediated decay transcripts *CHKB-CPT1B* and *SLX1B-SULT1A4* in ALS-TD patients may implicate a TDP-43 associated mechanism similar to those detailed in the process of cryptic exon splicing in *STMN2* and *UNC13A*^{100,101,102,103}. Should this be the case, nonsense-mediated decay of read-through genes *CHKB* and *CPT1B* suggests deficits in mitochondrial lipid metabolism, known to play a pathogenic role in muscular dystrophy¹⁰⁴, while loss of *SLX1B* and *SULT1A4* indicate impaired genome stability and monoamine synthesis. Similarly, increased expression of

retrotransposons, intronic, antisense, and long non-coding RNA implicates the TDP-43 protein in phenotypic presentation, given findings from Tam et al. and others^{7,99}. Of interest, both Brown et al.¹⁰¹ and Rosa Ma et al.¹⁰² report incomplete detection of the *UNC13A* cryptic exon in ALS patients from the same cohort¹⁰. Brown et al. observed that 38% of ALS patients expressed the cryptic exon in the *UNC13A* gene, while Rosa Ma et al. found 6.8% (31/454) of frontal and motor cortex samples had detectable levels of the *UNC13A* cryptic exon. In our analysis, we find that 26.9% of patients dominantly expressed the transcriptional dysregulation phenotype, potentially linking ALS-TD patients and cryptic exon expression in *UNC13A* – although additional work is needed to determine if cryptic exon expression is specific to ALS-TD. Despite our observation that *TARDBP* (encoding TDP-43) expression was relatively conserved across subtypes (Fig. S4B), our enrichment, WGCNA, and univariate results, suggest that TDP-43 pathological mechanisms, stemming from mislocalization, drive the expressed phenotype in ALS-TD patients. Given *TARDBP* expression in this cohort, we hypothesize subtype-specific differences in TDP-43 pathology occur at the protein level, and note support for this reasoning in the observation of TDP-43 hyperphosphorylation by Neumann et al.⁹⁷.”

97. Neumann, M. et al. Ubiquitinated TDP-43 in frontotemporal lobar degeneration and amyotrophic lateral sclerosis. *Science* **314**, 130-133 (2006).

98. Igaz, L.M. et al. Dysregulation of the ALS-associated gene TDP-43 leads to neuronal death and degeneration in mice. *The Journal of clinical investigation* **121**, 726-738 (2011).

99. Liu, E.Y. et al. Loss of nuclear TDP-43 is associated with decondensation of LINE retrotransposons. *Cell reports* **27**, 1409-1421 (2019).

100. Klim, J.R. et al. ALS-implicated protein TDP-43 sustains levels of STMN2, a mediator of motor neuron growth and repair. *Nature neuroscience* **22**, 167-179 (2019).

101. Brown, A.L. et al. TDP-43 loss and ALS-risk SNPs drive mis-splicing and depletion of *UNC13A*. *Nature* **603**, 131-137 (2022).

102. Ma, X.R. et al. TDP-43 represses cryptic exon inclusion in the FTD-ALS gene *UNC13A*. *Nature* **603**, 124-130 (2022).

103. Melamed, Z.E. et al. Premature polyadenylation-mediated loss of stathmin-2 is a hallmark of TDP-43-dependent neurodegeneration. *Nature neuroscience* **22**, 180-190 (2019).

104. Tavasoli, M. et al. Mechanism of action and therapeutic route for a muscular dystrophy caused by a genetic defect in lipid metabolism. *Nature communications* **13**, 1-20 (2022).

In Figure 3, the networks of pathways, edges denote sharing of genes. What does sharing with TEs mean? Is the TE network all a set of TEs rather than genes?

The authors thank the reviewer for their consideration of Fig. 3A. The sharing of network edges between the TE node and other pathways was a consequence of a small error in the way the custom enrichment file was generated. Using pattern-based string searching, the custom TE enrichment file contained two features (CR1 and MSR1) that were interpreted as gene symbols during edge construction. This minor error has been corrected for, and the TE node in the network no longer shares features with any other node.

This reviewer does not find dendrograms (3C) and network visualisations (3D-E) helpful. I would strongly suggest the authors rethink how they present their WGCNA network results.

The authors thank the reviewer for the recommended improvement in the presentation of WGCNA network results. Fig. 3 has been revised accordingly in the manuscript and reproduced below for convenience. Additional method details have been added to the manuscript in lines 36-41 on page 22. The original dendrogram and network visualizations (Fig. 3C-E) have been moved to the supplementary material as fig. S6.

Fig. 3. Network construction elucidates subtype-specific disease pathways and eigengenes associated with ALS patient clinical outcomes. (A) Network of pathways associated with the ALS cohort, color coded by subtype, with red indicating ALS-TD, blue denoting ALS-Ox, and yellow signifying ALS-Glia. **(B)** WGCNA identifies gene subsets significantly correlated with ALS patient age of disease onset, age of death, and disease duration. Eigengenes were enriched for gene ontology and Bonferroni-adjusted *p*-values are shown. Subtype-specific expression of eigengenes was determined using dummy regression, with the β coefficient presented as a heatmap. A positive β coefficient denotes subtype upregulation of transcripts comprising the particular eigengene. Bonferroni-adjusted *p*-values less than 0.05 are denoted with *. **(C)** Univariate plots showing gene expression levels of four features (*FCGR1B*, *FCGR3A*, *HLA-DOA*, *SERPINA3*) in the purple eigengene – with evidence for ALS-Glia specificity. **(D)** ALS-TD specific expression of four representative features (*ENSG00000215068*, *ENSG00000248015*, *KRT8P42*, *LINC00639*) in the magenta eigengene.

Reviewer 1 References

1. White, K . et al. Effect of postmortem interval and years in storage on RNA quality of tissue at a repository of the NIH NeuroBioBank. *Biopreservation and biobanking* **16**, 148-157 (2018).
2. Li, H . et al. The sequence alignment/map format and SAMtools. *Bioinformatics* **25**, 2078-2079 (2009).
3. Zhernakova, D.V. et al. Identification of context-dependent expression quantitative trait loci in whole blood. *Nature genetics* **49**, 139-145 (2017).
4. Nido, G.S. et al. Common gene expression signatures in Parkinson's disease are driven by changes in cell composition. *Acta neuropathologica communications* **8**, 1-4 (2020).
5. Giraldez, M.D. et al. Comprehensive multi-center assessment of small RNA-seq methods for quantitative miRNA profiling. *Nature biotechnology* **36**, 746-757 (2018).
6. Braak, H . et al. Amyotrophic lateral sclerosis—a model of corticofugal axonal spread. *Nature Reviews Neurology* **9**, 708-714 (2013).
7. Geevasinga, N ., Menon, P ., Özdinler, P.H., Kiernan, M.C., Vucic, S ., Pathophysiological and diagnostic implications of cortical dysfunction in ALS. *Nature Reviews Neurology* **12**, 651-661 (2016).
8. Brunet, A ., Stuart-Lopez, G ., Burg, T ., Scekcic-Zahirovic, J ., Rouaux, C ., Cortical circuit dysfunction as a potential driver of amyotrophic lateral sclerosis. *Frontiers in neuroscience* **14**, 363 (2020).
9. Fischer, L.R. et al. Amyotrophic lateral sclerosis is a distal axonopathy: evidence in mice and man. *Experimental neurology* **185**, 232-240 (2004).

Reviewer 2:

In the manuscript entitled “Molecular subtypes of ALS are associated with differences in patient prognosis”, Eshima et al. analyzed a large set of RNA-seq data from the frontal and motor cortices of ALS patients to distinguish molecular subtypes of ALS based on common gene expression signatures, which also included transposable elements (TEs) mapping uniquely to loci and quantified. They defined three ALS subtypes: ALS-Glia, ALS-Ox, and ALS-TD. Using Gene Set Enrichment Analysis to contrast each subtype’s expression profile against control cortices, they described some unique and some shared pathways among the subtypes. From this data set, they also constructed co-expression networks and highlighted modules comprised to subtype-specific genes. They developed a score-based classification approach based on these gene sets to classify ALS subtypes, and they refined the molecular subtypes of ALS present in their data set, demonstrated by observing that some samples were classified as hybrid subtypes. Finally, the authors assessed whether subtype classification through gene expression clustering could distinguish clinical features of the ALS patients. They demonstrated that the ALS-Glia subtype is significantly associated with a shorter survival duration.

Overall, this work applies several sophisticated methods to analyze transcriptomics data towards revealing insights into the heterogeneous nature of ALS. Like many consortia-based studies, this work aims to identify reliable biomarkers that can accurately diagnose and inform clinical decisions on ALS management and treatment. The noteworthy features of this manuscript are the large number of patient samples incorporated into the analyses and the management and reporting of the meta data. However, there are several concerns that limit the impact of the work overall.

This study seems to be an extension of the foundational work by Tam et al., 2019. <https://pubmed.ncbi.nlm.nih.gov/31665631/>, as the authors consistently reference. While many findings were reproduced, such as the classification of the 3 ALS subtypes and the gene expression pathways enriched in each, there are also different conclusions drawn. These include the observation that TARDBP is not reduced in the ALS-TD subtype (wherein Tam et al. the reduced expression of TARDBP explains the loss of repression of transposons) and the shorter survival of ALS-Glia subtypes (where Tam et al. observe no differences). How can the authors reconcile these different findings?

Different findings, in the context of *TARDBP* expression, are primarily driven by (i) the count normalization strategy employed by each group and (ii) sample size. Tam et al. provide counts on the read-per-million (RPM) scale while Eshima et al. consider counts on the DESeq2 median-of-ratios scale. While both methods account for the library size during normalization, the DESeq2 median-of-ratios scale has been shown to provide superior count estimates for between-sample comparisons¹. To demonstrate that our larger sample size explains some of these differences, we have provided a plot of *TARDBP* expression considering the Tam et al. cohort exclusively (Figure 10). Although we observe similar expression of *TARDBP* in both the ALS-Ox and ALS-TD subtypes, these findings are supported by locus-specific transposable element expression in this cohort (Fig. 2E, Fig. S5A, Fig. S12), given the associations between TDP-43 and transposons established by Tam et al.

TARDBP expression may be further driven by minor differences in sample classification, although we generally observe good agreement (85%, 119/140) for the majority of samples considered in the Tam et al. cohort (Table S11).

Figure 10. *TARDBP* expression in the Tam et al. cohort, presented on the median-of-ratios scale.

Figure 11. (A) ALS subtype survival in the Eshima et al. cohort. (B) ALS subtype survival in the Tam et al. cohort. No significant differences in survival are observed ($p > 0.1$).

We observe differences in patient survival, dependent on subtype, as a benefit of improved statistical power afforded by the larger patient cohort. To support this point, we have provided a “reduced” survival plot, excluding patients considered exclusively in the Prudencio et al. cohort (Figure 11). Similar to the findings reported by Tam et al., we do not observe significant differences in survival dependent on ALS subtype ($p > 0.1$, log-rank). Minor differences between the Tam et al. survival plot and the one provided in Figure 11 are driven by variability in patient clustering results during robust subtype estimation (see Materials and Methods & Tam et al. supplementary). Generally, we observe good agreement between our clustering results and Tam et al. findings (Supplemental Table 11).

While the data presented mostly supports the conclusions made by the authors, analyses could strengthen the support for their conclusions. Also, there are some flaws in the application of some analyses, as well as unclear descriptions of the methods (Details about this are explained in the specific comments on the figures and sections below).

Methods

Clustering: It is not clear if all or some of the 1475 TEs are included in the 10,000 most variable features used for clustering. Also, what is the extent of overlap of the 10,000 features between the HiSeq and Novaseq data sets?

Of the TEs passing the initial threshold detailed in the 'Methods' section, 1203 (81.6%) were included in the top 10,000 most variable features used for clustering in the NovaSeq cohort. In the HiSeq cohort, 644 (43.7%) of TEs were included in the top 10,000 most variable features. We observe that 97.7% (629/644) of TEs used for clustering in the HiSeq cohort were also used in the NovaSeq cohort, representing 42.7% (629/1474) of all TEs passing the initial threshold. In a more generalized context, 73.7% (7372/10000) of all features were shared between the NovaSeq and HiSeq cohorts. Some of this information has been added to the methods section, lines 9-17 on page 25.

Feature Selection: Why were the top 1,000 features selected from the 10,000 MAD transcripts? The remaining 1681 genes and TEs used for enrichment and network building are much lower than what is typically used for these analyses.

<https://www.ncbi.nlm.nih.gov/pmc/articles/PMC2756411/>

GSEA was performed on only 891 genes. For gene sets that are up to 500 genes (the default limit), which is over half of the 891 genes, this can likely lead to false enrichment. How was the ranked list generated?

A preliminary differential expression analysis considering platform-dependent gene expression found a large percentage of genes were differentially expressed after FDR-adjustment (37.2%, 22478/60403; including TEs). As a result, the authors elected to apply stringent filtering prior to enrichment and network analyses to limit sequencing platform biases. While this resulted in a smaller gene set considered during enrichment, the authors were focused on excluding features differentially expressed due to sequencing platform differences, rather than the underlying phenotype.

The rank listed was generated using the 'weighted' enrichment statistic, the 'Signal2Noise' metric with mean expression values, and the gene list sorted by the real metric score and in descending order. The 'meandiv' normalization mode was selected to provide normalized enrichment scores (NES). The 'fix metrics for low variance' option in the algorithm tab was left as default. As the reviewer notes, despite the smaller gene set used for enrichment and the default limit of 500 genes, we observe good agreement with Tam et al. with respect to the pathways enriched in each subtype. To demonstrate the default gene set limit did not influence the enrichment results, we reconsidered our analysis using a limit of 100 genes, with NES presented in Figure 12 of this response.

Figure 12. Normalized enrichment scores for subtype-specific pathway enrichment, with a gene set limit of 100 genes.

Classification: This section is not clear. Perhaps a description of an example predictor selection would help here. The R scripts on the Github repository are accessible, however the Rdata and csv input files are not.

The authors thank the reviewer for their recommendation to improve the method clarity for subtype predictor selection. We've modified the first paragraph of the 'Classification' section in an effort to enhance interpretability, lines 1-34 on page 23:

"Given that previously established predictor gene sets for ALS subtype were not available, ALS-Glia, ALS-Ox, and ALS-TD predictor gene sets were derived from our purple, turquoise, and magenta eigengenes, respectively (Fig. 3B). We utilized enrichment results from WGCNA and differential expression to establish subtype-specific expression of each eigengene. For example, most transcripts comprising the purple eigengene are specifically upregulated in the ALS-Glia subtype (Fig. 3C; fig. S6). Transcript counts were considered on the DESeq2 median-of-ratios scale, adjusted for RIN, site of collection, and sequencing platform covariates.

Subtype scores, defined as the average expression of subtype-specific predictor genes minus the average expression of all 1681 features considered in this analysis, were calculated for 100 different sets of predictors (per subtype) and used to define a 5% cutoff for the expected subtype score²¹. Each sampled predictor gene set contained the same number of features as the original eigengene, and were generated by randomly sampling the eigengenes with replacement. For example, the expected subtype score for ALS-Glia patients was determined by first generating 100 predictor sets by randomly sampling features comprising the purple eigengene. Then, the average (sample-wise) ALS-Glia expression was determined for each of the 100 predictor sets and subtracted from the average (sample-wise) ALS-Glia expression of all 1681 classification genes.

After repeating this analysis for the ALS-Ox and ALS-TD subtypes, using their respective eigengenes, 100 'subtype scores' were generated for all 451 samples. A 5% cutoff for the expected subtype score was then established, per sample, and final subtype classification thresholds were determined by weighting expected subtype scores according to the observed proportion of patient samples in each subtype (obtained from clustering). Bootstrapping was then applied, involving the sampling of predictor gene sets (with replacement) and calculation of subtype scores for 1000 iterations.

Patient samples were initially placed at the origin, and moved in the direction of the subtype vertex after passing the corresponding subtype threshold. Therefore, the x, y, and z axis vertices reflect the expression of a single subtype, while the other three vertices capture a combination of two subtypes. Individual points that passed a given subtype threshold in >50% of bootstrap iterations were filled with their respective subtype colors. Samples were considered to express a hybrid subtype state if one subtype threshold was passed >50% of the time and simultaneously passed a second subtype threshold >40% of the time."

As the reviewer notes, R and Python scripts to replicate this analysis, from the raw RNA-seq data files, are publicly available in the Github repository. CSV input files are provided as supplementary tables, where possible, although file size limitations and the large cohort size prevented the inclusion of all CSV files. For the same reason, RData files could not be provided, but these files have been made available at: https://figshare.com/authors/Jarrett_Eshima/13813720. Additionally, the publicly available code is commented to indicate the data source. The manuscript has been edited in lines 26-28 on page 25, to reflect this data availability.

Overall, the Figures are presented in low resolution and text was difficult to read.

The authors thank the reviewer for their recommendation. All figures have been edited to improve readability and updated accordingly in the manuscript and supplemental.

Figure 1: Do ALS/FTLD cases segregate?

FTLD patients were not included in the clustering analysis shown in manuscript Figure 1, as the primary goal was to stratify ALS patients to identify molecular subtypes, rather than assess potential markers differentiating the two neurodegenerative diseases. Furthermore, inclusion of confounding variables (i.e. sex-dependent genes, FTLN patients, sequencing platform, etc.) can limit the efficacy of non-negative matrix factorization, as shown in Figure 13.

Figure 13. nsNMF clustering in the NovaSeq cohort, including FTLN patients, using a rank of 4 and 200 iterations. (A) 100% of FTLN patients are classified as cluster 4 (green) and no ALS patients were included in this cluster, showing the two diseases separate well in the NovaSeq cohort. (B) Consensus matrix including all NovaSeq ALS (n=255) and FTLN (n=42) patient samples. All FTLN patient samples considered in this study were characterized using the NovaSeq platform.

Figure 2: A-C: Are these enriched gene sets with adjusted p -values at a > 0.05 threshold? Is this a comprehensive list of significant gene sets? D: to what extent do these TEs overlap with those analyzed by Tam et al.? Are these the same TEs bound by TDP43?

In an effort to demonstrate enrichment for the DAM microglia phenotype and other custom gene sets, results are presented as normalized enrichment score, rather than adjusted p -value. Many of the gene sets presented in Figs. 2A-C have a nominal p -value < 0.05 , although the smaller gene set used for enrichment leaves all adjusted p -values > 0.05 . To lend support to our GSEA results, the authors have paired differential expression with the Fisher's exact test² to assign enriched pathways to ALS subtype. The results are presented below as Figure 14 in this response, and added to Figure 2 in the manuscript as 2A. Benjamini-Hochberg adjusted p -values have been $-\log_{10}$ transformed and all presented pathways have adjusted p -values < 0.05 . Figure 14 contains a comprehensive list of gene sets enriched with an adjusted p -value < 0.05 , using the Fisher's exact test. The manuscript has been edited to reflect this additional analysis, in lines 4-5 on page 4, the caption for Fig. 2, and methods section, page 21 line 44 – page 22 line 4.

Figure 14. Hypergeometric enrichment analysis paired with differential expression assigns subtype-specific pathway expression. Benjamini-Hochberg adjusted p -values, derived from a Fisher's exact test, are presented on the $-\log_{10}$ scale. All presented pathways are significantly enriched in at least one subtype. Negative enrichment is encoded as the negative magnitude of the $-\log_{10}$ (adjusted p -value).

The TEs presented in Figure 2D are not directly comparable to the TE features considered by Tam et al. For the identification and quantification of transposons, Eshima et al. use SQulRE [Ref 13], whereas Tam et al. use Tetranscripts³. While both approaches utilize the EM algorithm for allocation of multi-mapping reads, TE ‘resolution’ differs. SQulRE offers locus-specific transposable element quantification, whereas Tetranscripts is limited to the subfamily level. Furthermore, Tetranscripts sums locus-level counts to provide subfamily level estimates, and as a consequence Tetranscripts introduces batch effects during quantification, supported by the hierarchical clustering analysis considering subfamily level co-expression (Table S7).

Most of the transposable element *superfamilies* shown to bind TDP-43 by Tam et al. (LINE, SINE, and ERV) are captured in Figure 2D (Table S6), excluding the SVA class. Retrotransposons belonging to the SVA class did not pass our initial TE filtering threshold (see Methods section; Table S3), again suggesting subfamily-level batch effects.

It is worth noting that many DNA classes shown to bind TDP-43 by Tam et al., other than TEs, were also observed to have ALS-TD specific expression including: long non-coding RNA, miRNAs, and intronic transcripts. Despite our observation that *TARDBP* expression is relatively conserved across ALS subtypes, the near ubiquitous nature of TDP-43 cellular inclusions in ALS patients clearly suggests a pathological role for this protein. Therefore, our results may suggest that subtype-specific differences in TDP-43 pathology occur at the protein level and implicate a mechanism other than transcription. This hypothesis is discussed further in the manuscript, on page 17 lines 45 – page 18 lines 28.

Figure 3: Break down each module by the percentage of TEs present in each. Could that explain why Magenta module does not enrich for gene ontologies?

The authors thank the reviewer for this observation regarding gene membership in the magenta module. The authors believe this point has already been addressed in the manuscript in lines 1-4 on page 7:

“The magenta eigengene – primarily composed of transposable elements, long non-coding RNA, pseudogenes, and poorly characterized transcripts (Ensembl IDs) – was not significantly linked to any gene ontologies, although a general association with transcription is perhaps a reasonable interpretation.”

Figure 4: The value of this analysis is low. NMF was already used to define the 3 clusters. From this, the modules were selected based on their associations with each subtype, and the classifiers were taken from the modules, and then reapplied to classify samples from the same data set. This is self-referential. Patel et al. <https://pubmed.ncbi.nlm.nih.gov/24925914/>, defined classifier genes from TCGA and applied these to classify inn new data sets of single cells in tumors. Despite this re-classification, what more can be said about the hybrid states phenotypically? Do they have altered survival, age of onset, etc? Visually, it is difficult to distinguish border colors from fill colors on the plot. Consider showing histograms and quantification of the classifications.

The authors thank the reviewer for their consideration of Figure 4.

The aim of this analysis was to assess whether the subtypes identified by NMF can be co-expressed (hybrid subtypes). Unsupervised clustering is inherently limited in addressing this question, given samples are assigned to a single cluster. Our robust subtype estimation approach (see Methods), begins to broach the question of hybrid subtypes, and we used our observation of 'edge' cases (see Methods) as supporting evidence for performing this bootstrap-based classification.

The authors agree with the reviewer that the selection of ALS subtype predictor gene sets would be best derived from an independent source or dataset. As the reviewer notes, Patel et al. define tumor subtype classifier genes from TCGA. No such reference gene sets exist for ALS subtypes, as a consequence the authors leveraged a data-driven approach (subtype-specific eigengenes).

The authors thank the reviewer for their recommendation and have performed additional analyses to further consider the consequences of the hybrid subtype states from a clinical perspective. Figure 4 has been modified in the manuscript to improve interpretability, and a new supplemental figure was added to address clinical outcomes in the context of hybrid subtypes (Figure S8) and referenced in the manuscript in lines 7-8 on page 10. Both figures are reproduced below for convenience.

Fig. 4. Score-based classification uncovers hybrid subtype states in the ALS cohort. (A) Subtype scoring was implemented with bootstrapping to assess the spectrum of disease phenotypes presented in ALS. Each point corresponds to a single transcriptome derived from the frontal or motor postmortem cortex. Patient samples were initially placed at the origin, moved in the direction of the subtype axis for each round of bootstrapping that passed the subtype score threshold, and could only reach the vertex if the patient sample passed the threshold in all rounds of bootstrapping. Data points are filled according to the bootstrap-based subtype assignment and borders are included to denote the patient subtype obtained from unsupervised clustering. Transcriptomes which approached the vertices shared by two subtypes are considered to express a hybrid subtype state. Patient samples are color coded gray if they failed to pass the subtype score thresholds in $\geq 50\%$ of bootstrap iterations. **(B)** Confusion matrix showing unsupervised clustering results in each classification subtype. **(C)** Clustering results in Glia-TD and Glia-Ox hybrids.

Fig. S8. Clinical parameters in the hybrid subtypes. Patients were assigned to a hybrid subtype if one or more tissue samples passed the thresholds detailed in the ‘Methods’ section. (A) Kaplan-Meier survival analysis including the three ALS subtypes, hybrids, and ‘discordant’ patients. Interestingly, survival in Glia-TD hybrids mirrors survival in the ALS-Glia subtype, with significant differences observed when compared to the ALS-TD subtype ($p = 0.007$), and survival differences trending towards significance when compared to the ALS-Ox subtype ($p = 0.085$). Our findings suggest the elevated inflammatory phenotype seen in ALS-Glia patients is sufficient to drive fast progression in ALS, irrespective of co-expressed phenotypes, although additional work is needed to assess the consistency of hybrid subtype expression in other cohorts. (B) Age of symptom onset, plotted as boxplots, and separated by subtype. No significant differences are observed between the Glia-TD hybrids and other subtypes. (C) Age of death, separated by subtype. (D) Site of symptom onset for all disease subtypes. (E) FTL D comorbidity in each disease subtype, presented as a percentage. The small number of Glia-Ox hybrids limits the interpretation of differences observed in survival, age of onset ($p < 0.05$ for all pairwise comparisons), age of death ($p < 0.05$ for all pairwise comparisons), and FTL D comorbidity.

Figure 5: A: the outcome of this analysis is different from Tam et al. This is worth a mention and analysis in the Discussion section.

The authors thank the reviewer for their recommendation. The following sentences have been added to the discussion section to address this point, lines 35-45 on page 16:

“Noteworthy findings, differing from the foundational Tam et al. study, include (i) our redefinition of the transposable element subtype – driven primarily by our consideration of transposable elements at the gene locus level, (ii) our identification of an immunological eigengene significantly correlated with age of disease onset and survival, and (iii) our observation that the ALS-Glia subtype is associated with a significantly shorter survival duration. However, despite the redefinition of the transposable element subtype, we generally observe good agreement with Tam et al. with respect to the major pathological themes identified in each subtype. Furthermore, given Tam et al. demonstrate TDP-43 binds and regulates a variety of non-protein coding genes including intronic, long non-coding and regulatory RNA, transposable elements, and intergenic DNA our results suggests TDP-43 plays a core role in the ALS-TD phenotype.”

Figure 6: FTLD comorbidity ALS cases should be analyzed separately if they are to be compared to FTLD cases.

The authors thank the reviewer for the recommended analysis in the context of manuscript Figure 6. As seen in manuscript Fig. 5E, less than 20% of patients, per subtype, were co-morbid for FTLD in this cohort. The significantly reduced sample size, in combination with our results from the χ^2 test of independence, led us to include ALS patients with and without FTLD in this analysis.

The authors performed the recommended analysis in an effort to demonstrate disease specificity for the features considered in manuscript Figs. 6 and S13. Figure 16 from this response have been included as a new supplemental figure (S14) and referenced in the manuscript in lines 17-20 on page 11.

Figure 15. Features presented in manuscript Fig. 6 are reconsidered, excluding FTLD–ALS patients. Subtype-specific expression is maintained in this reduced cohort, and some features (*HLA-DRA*, *TLR7*, *AGPAT4-IT1*, *CHKB-CPT1B*, *HSP90AB4P*, *miR24-2*, and *ENSG00000205041*) reveal stark differences in expression between ALS-FTLD and FTLD patients, suggesting these features are specific to ALS pathology.

Figure 16. Features presented in manuscript Fig. S13 are reconsidered, excluding ALS patients without FTLD. Differential expression between ALS-FTLD and FTLD patients is maintained, again suggesting these features are specific to ALS pathology.

Figure S5: B: Would the results be the same for the 142 overlapping samples in Tam et al.? How do you reconcile the different findings? Were the same TEs quantified between the two studies? Is it due to platform or VST differences?

This question is previously discussed on pages 21 and 28 of this response, but is provided below for reference.

Different findings, in the context of *TARDBP* expression, are primarily driven by (i) the count normalization strategy employed by each group and (ii) sample size. Tam et al. provide counts on the read-per-million (RPM) scale while Eshima et al. consider counts on the DESeq2 median-of-ratios scale. To demonstrate that our larger sample size explains some of these differences, we have provided a plot of *TARDBP* expression considering the Tam et al. cohort exclusively (Figure 11). Although we observe similar expression of *TARDBP* in both the ALS-Ox and ALS-TD subtypes, these findings are supported by locus-specific transposable element expression in this cohort (Fig. 2E, Fig. S5A, Fig. S12), given the associations between TDP-43 and transposons established by Tam et al.

TARDBP expression may be further driven by minor differences in sample classification, although we generally observe good agreement (85%, 119/140) for the majority of samples considered in the Tam et al. cohort (Table S11). We also note that the Prudencio et al. use both NovaSeq and HiSeq platforms, while Tam et al. use HiSeq exclusively.

Figure 10. *TARDBP* expression in the Tam et al. cohort, presented on the median-of-ratios scale.

Similarly, as discussed on page 28 of this response, the locus-specific TEs considered in our analysis are not directly comparable to the TE features considered by Tam et al. due to differences in quantification methodologies. Our hierarchical clustering analysis considering subfamily level co-expression (Table S7) supports the claim that SQuIRE is better suited to quantify TE features in this cohort.

Figure S6: B-E: Consider performing Wilcoxon Rank sum to assess significance of prediction accuracy.

A Wilcoxon rank sum test was applied in a pair-wise manner to assess significant differences in prediction accuracy amongst the four classifiers, using AUC as the response metric. No significant differences in prediction accuracy were observed.

The authors also approached this question using net reclassification improvement and integrated discrimination improvement (IDI) methodology detailed by Pencina et al.⁴. Classification models were assessed in a pairwise manner, for each instance of the one-vs-rest classifier (Table 4). Two classifiers were determined to perform equivalently if the integrated discrimination improvement 95% confidence interval included 0. In the Glia-vs-rest case, no single classifier was observed to outperform the others. For models considering the Ox-vs-rest case, both the random forest and linear support vector classifiers were seen to outperform the multilayer perceptron and k-nearest neighbor classifiers. Overall, the SVM classifier outperformed all others in the prediction of ALS-Ox cases versus other subtypes. Finally, in the TD-vs-rest case, both the MLP and SVM classifiers outperformed the RF classifier, although no classifier was clearly best at predicting ALS-TD patients versus other subtypes.

These findings have been added to the figure caption on page 10 of the supplemental materials.

ALS-Glia vs. Rest			
Classifier 1	Classifier 2	IDI Lower 95% CI	IDI Upper 95% CI
MLP	KNN	-0.0623	0.0958
MLP	RF	-0.0316	0.0960
MLP	SVM	-0.0290	0.0836
RF	KNN	-0.0878	0.0569
RF	SVM	-0.0279	0.0182
SVM	KNN	-0.0816	0.0605
ALS-Ox vs. Rest			
Classifier 1	Classifier 2	IDI Lower 95% CI	IDI Upper 95% CI
MLP	KNN	-0.0061	0.0574
MLP	RF	-0.0822	-0.0098
MLP	SVM	-0.1270	-0.0555
RF	KNN	0.0376	0.1060
RF	SVM	-0.0713	-0.0189
SVM	KNN	0.0783	0.1550
ALS-TD vs. Rest			
Classifier 1	Classifier 2	IDI Lower 95% CI	IDI Upper 95% CI
MLP	KNN	-0.0098	0.0380
MLP	RF	0.0061	0.0910
MLP	SVM	-0.0439	0.0524
RF	KNN	-0.0709	0.0020
RF	SVM	-0.0705	-0.0181
SVM	KNN	-0.0347	0.0544

Table 3. Integrated discrimination improvement 95% confidence intervals for each instance of the one-vs-rest classifier. A strictly positive 95% CI indicates classifier 1 has superior prediction accuracy, while a strictly negative 95% CI shows classifier 2 is better suited for prediction.

Figure S7: A: Is it fair to include survival curves for subjects with only 1 tissue profiled? These patients are classified with lower confidence than those patients with frontal and motor cortices both classified as the same subtype.

Our results show multiple tissue samples from the same subject often are classified as the same subtype (80.8%; 126/156; Table S10). Therefore, despite the lower confidence, our findings suggest that the majority of patients with a single tissue profiled would retain their subtype label, supporting the inclusion of all ALS patients during the survival analysis. The following sentence has been added to the manuscript in lines 25-27 on page 9 to address this point:

“Importantly, we observe that multiple tissue samples from the same donor are classified as the same subtype (80.8%; 126/156), lending support to our subtype assignment methodology.”

Figure S12: If these are strong candidates as markers for distinguishing FTL and ALS/FTL, consider supporting this idea by testing them as predictors in a validation data set.

To further consider candidate markers distinguishing FTL and ALS cases, we performed additional supervised classification analyses, using all features presented in Figure S12 and transcripts *AIF1*, *APOC2*, *HLA-DRA*, *AGPAT4-IT1*, *CHKB-CPT1B*, *HSP90AB4P*, *MIR24-2*, *FCGR3A*, *ADAT3*, *LINC00176*, *MIR219A2*, *SLX1B-SULT1A4*, *TUB-AS1* from Figures 3, 6 and S12 – totaling 33 features. All FTL patients had frontal and motor postmortem transcriptomes characterized on the NovaSeq platform and, as a consequence, classifier performance in the ‘validation’ HiSeq cohort could not be assessed. Expression counts were VST transformed prior to classifier development. All four classifiers demonstrated perfect classification accuracy across all 100 rounds of cross-validation and representative ROC plots are shown in Figure 18. Although classifier training metrics tend to overestimate the predictive accuracy when applied independent data sets, our results emphasize the strength of the candidate markers for distinguishing FTL and ALS.

Figure 17. Representative ROC plots showing sensitivity (TPR) and 1-specificity (FPR) for the binary classification problem involving ALS and FTL patients. The test dataset was generated by randomly sampling 30% of patient samples from the NovaSeq cohort (n=297, ALS and FTL). Classifier performances were assessed over 100 rounds of cross validation.

Reviewer 2 References

1. Zhao, Y. et al. TPM, FPKM, or normalized counts? A comparative study of quantification measures for the analysis of RNA-seq data from the NCI patient-derived models repository. *Journal of translational medicine* **19**, 1-5 (2021).
2. Kuleshov, M.V. et al. Enrichr: a comprehensive gene set enrichment analysis web server 2016 update. *Nucleic acids research* **44**, W90-W97 (2016).
3. Jin, Y., Tam, O.H., Paniagua, E., Hammell, M., Tetranscripts: a package for including transposable elements in differential expression analysis of RNA-seq datasets. *Bioinformatics* **31**, 3593-3599 (2015).
4. Pencina, M.J., D'Agostino, R.B., Sr., D'Agostino, R.B. Jr., Vasan, R.S., Evaluating the added predictive ability of a new marker: from area under the ROC curve to reclassification and beyond. *Statistics in medicine* **27**, 157-172 (2008).

Reviewer 3:

Eshima et al. address the question of heterogeneity in ALS by reanalyzing bulk RNAseq data from the motor/frontal cortex of over 200 patients and performing a patient stratification analysis. They identified 3 molecular subtypes, ALS- Glia (glia activation); ALS-Ox (oxidative stress and altered synaptic signaling), and ALS-TD (transcriptional dysregulation), and linked them to the age of onset and survival as clinical outcomes. Although interesting, there are concerns regarding the methodology used to identify subtypes and their clinical relevance:

1) The question addressed is very complex, however the paper does not consider some important factors, both clinical/biological metadata as well as potential technical artifacts, that can affect RNAseq data. For instance:

- Region: The authors classify the regions used as “frontal” and “motor” cortices; what region(s) of the frontal lobe were used? Different frontal regions have different cellular compositions and are not equally affected in ALS (or in cohort of FTLN patients used as controls).

- Although data from different brain regions are available from this cohort, the authors used only frontal/motor. Including “control” regions that are not affected in ALS such as the cerebellum is helpful to address covariates such as the RIN, the postmortem interval, agonal stage, and genetic factors

- Clinical and pathological heterogeneity is described in ALS, how do the three subtypes identified by this paper relate to those?

Region: The authors classify the regions used as “frontal” and “motor” cortices; what region(s) of the frontal lobe were used? Different frontal regions have different cellular compositions and are not equally affected in ALS (or in cohort of FTLN patients used as controls).

Patient samples were assigned tissue region by the New York Genome Center and Target ALS and are reported as provided. We observe approximately the same ratio of Glia, Ox and TD patients in the frontal and specified motor cortices, roughly matching the ratio observed during unsupervised clustering, indicating brain region is not a confounding factor with subtype in our analysis. Figure 3 from this response has been added as a new supplementary figure (Figure S16A) and referenced in the manuscript in lines 16-26 on page 15.

Figure 3. Patient brain region parsed by subtype

To further address concerns that cortex region cellular composition may explain some subtype-specific differences in expression, we utilized CIBERSORT to assess cell type percentages in the two regions.

Figure 18. Cell type percentages in the prefrontal and motor cortices for all patient samples considered in this study. Roughly the same proportion of cell types are observed in each tissue, although significant differences are observed for microglial, glial progenitor, vascular, and inhibitory neurons. Significant

differences in cell type percentages were assessed using a Wilcoxon rank sum test with Bonferroni p -value adjustment. Adjusted p -values are denoted using the following scheme: *** $p < 0.001$; ** $p < 0.01$; * $p < 0.05$.

Our results, presented in Figure 18 of this response, show significant differences in microglial, glial progenitor, vascular cells, and inhibitory neuron fractions. No significant differences in astrocyte fractions are observed and only weakly significant differences in excitatory neuron percentages are reported between the two regions. Taken together, these findings indicate cell percentages in the frontal and motor cortex may partially explain subtype-specific expression, although Figure 3 clearly shows tissue region biases do not strongly influence our assignment of ALS subtype. Figure 18 from this response has been added to the manuscript as Fig. 7A and discussed in the text in lines 21-28 on page 15.

Although data from different brain regions are available from this cohort, the authors used only frontal/motor. Including “control” regions that are not affected in ALS such as the cerebellum is helpful to address covariates such as the RIN, the postmortem interval, agonal stage, and genetic factors

The comprehensive use of control regions, such as the cerebellum, to address covariates such as postmortem interval and agonal stage, was not feasible in this cohort, given that only 150 out of the 208 unique ALS patients had a corresponding cerebellum sample characterized. Differential expression results have been adjusted for RIN and site of collection covariates (Table 2) and sample-wise RIN values have been added to Table S2. We further considered genetic factors in this cohort using the Chi-squared test of independence and results suggest C9orf72 and SOD1 mutations are not associated with ALS subtype (Figure 7).

Design 1: sequencing platform + subtype

Design 2: sequencing platform + RIN + site + subtype

Gene	Design 1: Glia vs Ox	Design 2: Glia vs Ox	Design 1: Glia vs TD	Design 2: Glia vs TD	Design 1: Ox vs TD	Design 2: Ox vs TD
AIF1	6.95E-15	3.08E-14	2.58E-23	8.20E-19	2.92E-04	3.72E-02
APOC2	1.71E-10	6.91E-12	6.38E-06	5.08E-06	1.89E-01	6.92E-02
CD44	4.55E-17	1.36E-17	1.51E-10	2.69E-10	2.10E-01	1.16E-01
CHI3L2	1.02E-17	5.96E-18	9.34E-12	3.29E-11	3.51E-01	2.21E-01
CX3CR1	8.26E-04	9.52E-03	1.00E-10	1.84E-07	2.15E-05	1.16E-03
FOLH1	3.25E-06	5.01E-07	7.16E-05	3.84E-06	8.92E-01	1.00E+00
HLA-DRA	2.86E-16	5.58E-17	1.77E-22	8.85E-19	3.84E-03	2.40E-01
TLR7	2.60E-08	5.05E-09	1.19E-07	2.39E-07	7.92E-01	9.40E-01
TMEM125	7.49E-10	6.41E-10	3.59E-10	1.93E-11	4.45E-01	2.88E-01
TNC	3.49E-16	1.08E-14	1.24E-09	1.31E-10	1.64E-01	5.90E-01
TREM2	1.71E-18	7.11E-18	3.56E-09	1.32E-10	1.99E-02	1.27E-01
TYROBP	1.46E-24	1.59E-24	1.98E-17	8.90E-18	4.79E-01	4.69E-01
COL18A1	4.63E-44	3.49E-35	3.17E-04	2.97E-08	2.59E-30	1.37E-12
GABRA1	1.01E-10	1.91E-13	6.80E-02	6.53E-01	1.00E-24	3.54E-14
GAD2	3.84E-03	4.08E-06	7.63E-05	5.50E-02	6.95E-19	1.64E-15
GLRA3	1.12E-11	1.18E-11	9.51E-01	2.32E-01	2.28E-16	3.27E-09
HTR2A	8.31E-24	1.50E-26	1.99E-01	9.59E-05	9.55E-23	1.30E-12
OXR1	1.82E-15	2.32E-17	6.28E-02	6.35E-01	2.38E-33	1.20E-18
SERPINI1	7.05E-06	2.16E-09	3.30E-07	2.69E-02	4.74E-34	7.36E-23
SLC6A13	9.40E-29	1.33E-12	1.90E-05	2.34E-04	5.77E-13	1.46E-03
SLC17A6	5.91E-09	4.95E-07	3.98E-01	9.29E-01	2.59E-16	4.96E-08
TCIRG1	1.40E-28	4.17E-23	5.54E-02	1.57E-04	3.16E-24	2.64E-10
UBQLN2	1.86E-04	1.23E-06	9.25E-04	4.74E-02	1.98E-19	1.22E-16
UCP2	8.33E-21	1.90E-23	1.55E-10	1.73E-13	1.85E-02	4.97E-02
AGPAT4-IT1	1.56E-07	1.93E-08	1.23E-04	2.92E-01	4.01E-30	1.20E-15
CHKB-CPT1B	1.89E-07	1.05E-05	2.54E-15	5.51E-10	3.07E-61	9.84E-38
COL3A1	3.13E-18	1.50E-07	2.19E-13	2.20E-09	6.44E-01	2.04E-01
ENSG00000205041	3.82E-01	7.10E-01	2.69E-30	2.61E-22	2.81E-59	3.25E-37
ENSG00000258674	1.64E-01	4.32E-01	8.05E-25	5.69E-17	3.76E-53	2.86E-31
ENSG00000273151	2.06E-09	2.05E-09	4.45E-06	3.82E-02	9.21E-40	3.23E-22
GATA2-AS1	8.94E-07	1.27E-04	3.03E-08	6.50E-05	2.69E-39	1.31E-21
HSP90AB4P	3.24E-04	2.22E-06	1.14E-16	4.03E-07	4.90E-52	1.77E-32
LINC01347	5.08E-03	1.72E-02	5.41E-21	1.05E-15	1.03E-55	4.74E-38
MIR24-2	1.40E-10	4.06E-07	9.28E-05	3.62E-03	1.20E-37	5.60E-22
MIRLET7BHG	9.33E-15	5.45E-12	4.19E-19	6.75E-12	4.39E-94	4.05E-60
NANOGP4	1.63E-07	9.25E-10	2.56E-02	7.42E-01	1.34E-20	2.69E-10

Table 2. FDR-adjusted *p*-values comparing the original design equation and one accounting for both site and RIN as covariates. No major differences in DE significance were observed for the 36 subtype-specific features.

Figure 7. Stacked bar chart showing C9orf72 and SOD1 mutation frequency in the ALS cohort. A chi-squared test of independence was performed to assess mutation dependency on subtype. After removal of the “unknown” categorical variable, the null hypothesis (no association between ALS subtype and common genetic drivers) was accepted for both C9orf72 ($p = 0.47$) and SOD1 ($p = 0.21$). It is important to note that the limited number of observations for SOD1 may drive inaccurate estimation of the chi-squared test statistic.

The following changes to the manuscript have been made to reflect our adjustment for RIN and site of sample collection covariates during differential expression:

-lines 7-9 on page 23

-lines 37-39 on page 24

In addition, our consideration of genetic risk factors in the context of the identified subtypes has been added as a new supplemental figure (fig. S15E) and discussed in the text in lines 20-23 on page 11.

Clinical and pathological heterogeneity is described in ALS, how do the three subtypes identified by this paper relate to those?

The authors thank the reviewer for their recommendation to consider these subtypes in the context of previous works considering clinical and pathological heterogeneity. The following sentences have been added to the discussion section of the manuscript to address this point, page 18 line 30 – page 19 line 11.

“Clinical and pathological heterogeneity are well established features of Amyotrophic Lateral Sclerosis. Heterogeneity in clinical presentation is typically characterized by region of onset, mixture of upper and lower motor neuron involvement, and rate of progression¹⁰² – although this scheme often fails to accurately predict patient outcomes^{3,4}. As a consequence of the poorly understood clinical heterogeneity in ALS, significant research efforts aimed at unraveling the molecular underpinnings in patients^{7,8,9} and animal and cell models^{28,38,54,58,59,103} have implicated a number of disease mechanisms shown to contribute to pathological variability. As detailed by Taylor, Brown Jr., and Cleveland¹², these mechanisms include (1) disturbances in protein quality control including autophagy, proteasome-mediated degradation, and endosome-lysosome mediated degradation, (2) hyperactivated microglia, (3) decreased energy supply from oligodendrocytes following downregulation of MCT1, (4) glutamate excitotoxicity, (5) disturbances in RNA metabolism, and (6) cytoskeletal defects and altered axonal transport. Importantly, the three subtypes identified in this work directly capture the majority of these proposed mechanisms. Supported by our differential expression results, disturbances to protein quality control (proteotoxic stress) is a defining hallmark of ALS-Ox patients, while the hallmark of the ALS-TD patient phenotype is dysregulated RNA metabolism. In ALS-Glia patients, we observe upregulation of inflammatory genes, implicating activated microglia and astrocytes in the accelerated progression of disease pathology. Beyond the major pathological themes of each subtype, we observe some evidence for cytoskeletal defects and altered axonal transport in ALS-Ox and ALS-TD patients through expression of *ACTA2*, *DYNLT3*, *PLS1*, and *TUBB6*. Moderate overexpression of *FOLH1* implicates glutamate excitotoxicity in ALS-Glia patients, although further consideration of transcripts encoding glutamate receptors and glutamate synthesizing enzymes is needed to explore subtype specificity. In summary, this work helps to clarify the molecular foundation of clinical and pathological heterogeneity in ALS by demonstrating subtype-specific phenotypes are associated with patient outcomes, including survival and age of onset.”

102. Ravits, J.M., La Spada, A.R., ALS motor phenotype heterogeneity, focality, and spread: deconstructing motor neuron degeneration. *Neurology* **73**, 805-811 (2009).

103. Strong, M.J., The evidence for altered RNA metabolism in amyotrophic lateral sclerosis (ALS). *Journal of the neurological sciences* **288**, 1-2 (2010).

2) Cellular composition is usually a driver of clustering in the analysis of bulk RNAseq, and so clusters unbiasedly identified may reflect the relative abundant of specific cell types. Cell type deconvolution methods to infer cell type proportions may help address this. This is important in this paper since two of the subtypes of ALS identified correspond with glial (ALS-Glia) and neuronal (ALS-Ox) responses.

The authors thank the reviewer for the recommendation to consider the effects of cell type proportion in our analysis of bulk tissue RNA-seq. The authors have performed cell deconvolution using CIBERSORT⁷⁷, with DESeq2 normalized count values and reference single cell expression from Nowakowski et al.⁷⁸. In an effort to ensure a sufficient number of features were available for deconvolution, the authors used all overlapping MAD transcripts between the NovaSeq and HiSeq cohorts, totaling 7372 transcripts. Removal of transcripts without a corresponding gene symbol and transposable elements led to 4912 transcripts, 1881 of which were shared between the ALS cohort and Nowakowski single cell reference expression.

Figure 6. Bulk tissue RNA-seq cell deconvolution in ALS subtypes. Fractions of cell types in the frontal and motor postmortem cortex, considered in the context of the ALS subtypes. Significant differences in cell type percentages was assessed using the Wilcoxon rank sum test with Bonferroni p -value adjustment. Adjusted p -values are denoted using the following scheme: *** $p < 0.001$; ** $p < 0.01$; * $p < 0.05$.

Generally, we observe some significant differences in cell type percentages between the three subtypes. In the excitatory and inhibitory neuronal populations, the ALS-Ox subtype had significantly greater percentage of neurons, as compared to the ALS-Glia subtype. Weakly significant differences in inhibitory cell percent are seen between ALS-TD and ALS-Ox patients, yet these differences are not significant in the excitatory neuron population estimate. These findings suggest bulk tissue biases capture some of ALS-Ox specific expression, although it is clear that varying cell type percentages do not fully explain characteristic expression differentiating Ox and TD subtypes. Similarly, cell deconvolution demonstrates ALS-Glia patient samples had a greater estimated fraction of microglial cells and glial progenitor cells, as compared to the other two subtypes. Despite the greater percentage of microglial and glial progenitor cells

in ALS-Glia samples, the ALS-Glia subtype did not demonstrate significantly different percentages of astrocytes, maintaining upregulated neuroinflammation as a defining characteristic.

The manuscript has been updated to include our cell deconvolution analysis, with the results presented on page 15 and 16. The methods section has been updated on page 25 – to describe our approach to this analysis.

3) Differential gene expression needs to take into account potential covariates beyond sex.

To account for other covariates beyond sex, we re-performed the differential expression analysis accounting for RIN, site of sample collection (NYGC vs Target ALS), and sequencing platform covariates. Our results demonstrate that these covariates do not fully explain the subtype-specific expression observed in this cohort, supporting the use of these features to stratify ALS patients based on molecular phenotype. Updated adjusted p -values have been summarized in Table 2 of this response, for the 36 features presented in manuscript Figure 6. We have elected to use median-of-ratios count values adjusted for RIN, site, and platform covariates and present these updated FDR p -values in Fig. 6, figs. S9-S14 from the manuscript. Table S12 has been updated with FDR-adjusted p -values accounting for RIN and site covariates for all pairwise comparisons. Additional method details have been added to the manuscript in lines 37-39 on page 24.

Original Design: sequencing platform + subtype

Updated Design: sequencing platform + RIN + site + subtype

Gene	Design 1: Glia vs Ox	Design 2: Glia vs Ox	Design 1: Glia vs TD	Design 2: Glia vs TD	Design 1: Ox vs TD	Design 2: Ox vs TD
AIF1	6.95E-15	3.08E-14	2.58E-23	8.20E-19	2.92E-04	3.72E-02
APOC2	1.71E-10	6.91E-12	6.38E-06	5.08E-06	1.89E-01	6.92E-02
CD44	4.55E-17	1.36E-17	1.51E-10	2.69E-10	2.10E-01	1.16E-01
CHI3L2	1.02E-17	5.96E-18	9.34E-12	3.29E-11	3.51E-01	2.21E-01
CX3CR1	8.26E-04	9.52E-03	1.00E-10	1.84E-07	2.15E-05	1.16E-03
FOLH1	3.25E-06	5.01E-07	7.16E-05	3.84E-06	8.92E-01	1.00E+00
HLA-DRA	2.86E-16	5.58E-17	1.77E-22	8.85E-19	3.84E-03	2.40E-01
TLR7	2.60E-08	5.05E-09	1.19E-07	2.39E-07	7.92E-01	9.40E-01
TMEM125	7.49E-10	6.41E-10	3.59E-10	1.93E-11	4.45E-01	2.88E-01
TNC	3.49E-16	1.08E-14	1.24E-09	1.31E-10	1.64E-01	5.90E-01
TREM2	1.71E-18	7.11E-18	3.56E-09	1.32E-10	1.99E-02	1.27E-01
TYROBP	1.46E-24	1.59E-24	1.98E-17	8.90E-18	4.79E-01	4.69E-01
COL18A1	4.63E-44	3.49E-35	3.17E-04	2.97E-08	2.59E-30	1.37E-12
GABRA1	1.01E-10	1.91E-13	6.80E-02	6.53E-01	1.00E-24	3.54E-14
GAD2	3.84E-03	4.08E-06	7.63E-05	5.50E-02	6.95E-19	1.64E-15
GLRA3	1.12E-11	1.18E-11	9.51E-01	2.32E-01	2.28E-16	3.27E-09
HTR2A	8.31E-24	1.50E-26	1.99E-01	9.59E-05	9.55E-23	1.30E-12
OXR1	1.82E-15	2.32E-17	6.28E-02	6.35E-01	2.38E-33	1.20E-18
SERPINI1	7.05E-06	2.16E-09	3.30E-07	2.69E-02	4.74E-34	7.36E-23
SLC6A13	9.40E-29	1.33E-12	1.90E-05	2.34E-04	5.77E-13	1.46E-03
SLC17A6	5.91E-09	4.95E-07	3.98E-01	9.29E-01	2.59E-16	4.96E-08
TCIRG1	1.40E-28	4.17E-23	5.54E-02	1.57E-04	3.16E-24	2.64E-10
UBQLN2	1.86E-04	1.23E-06	9.25E-04	4.74E-02	1.98E-19	1.22E-16
UCP2	8.33E-21	1.90E-23	1.55E-10	1.73E-13	1.85E-02	4.97E-02
AGPAT4-IT1	1.56E-07	1.93E-08	1.23E-04	2.92E-01	4.01E-30	1.20E-15
CHKB-CPT1B	1.89E-07	1.05E-05	2.54E-15	5.51E-10	3.07E-61	9.84E-38
COL3A1	3.13E-18	1.50E-07	2.19E-13	2.20E-09	6.44E-01	2.04E-01
ENSG00000205041	3.82E-01	7.10E-01	2.69E-30	2.61E-22	2.81E-59	3.25E-37
ENSG00000258674	1.64E-01	4.32E-01	8.05E-25	5.69E-17	3.76E-53	2.86E-31
ENSG00000273151	2.06E-09	2.05E-09	4.45E-06	3.82E-02	9.21E-40	3.23E-22
GATA2-AS1	8.94E-07	1.27E-04	3.03E-08	6.50E-05	2.69E-39	1.31E-21
HSP90AB4P	3.24E-04	2.22E-06	1.14E-16	4.03E-07	4.90E-52	1.77E-32
LINC01347	5.08E-03	1.72E-02	5.41E-21	1.05E-15	1.03E-55	4.74E-38
MIR24-2	1.40E-10	4.06E-07	9.28E-05	3.62E-03	1.20E-37	5.60E-22
MIRLET7BHG	9.33E-15	5.45E-12	4.19E-19	6.75E-12	4.39E-94	4.05E-60
NANOGP4	1.63E-07	9.25E-10	2.56E-02	7.42E-01	1.34E-20	2.69E-10

Table 2. FDR-adjusted p -values comparing the original design equation and one accounting for both site and RIN as covariates. No major differences in DE significance were observed for the 36 subtype-specific features.

REVIEWERS' COMMENTS

Reviewer #1 (Remarks to the Author):

I am satisfied that the authors have responded to all my concerns. The manuscript has been strengthened and I believe it is ready for publication. I applaud the authors on their hard work in developing these important insights for the ALS field.

Jack Humphrey
Icahn School of Medicine at Mount Sinai
New York City

Reviewer #2 (Remarks to the Author):

The authors have addressed all my major and minor comments.

In the new Figure 3B, the module colors should be labeled for clarity.

Reviewer #3 (Remarks to the Author):

The authors have performed a thorough revision of the original manuscripts and incorporated this reviewer's suggestions. Particularly, the question of potential confounders (brain regions, cellular composition heterogeneity, RIN, genetics, etc) has been addressed.

Response to Reviewers' Comments:

The authors would like to thank the reviewers for their further consideration of our revised work and interest in our findings. The authors have addressed all suggestions for improvement, and the following changes have been made to the manuscript text:

Figures:

- Fig. 3B – Module colors have been defined in the figure caption for added clarity.

Reviewer 1:

I am satisfied that the authors have responded to all my concerns. The manuscript has been strengthened and I believe it is ready for publication. I applaud the authors on their hard work in developing these important insights for the ALS field.

Jack Humphrey
Icahn School of Medicine at Mount Sinai
New York City

The authors thank the reviewer for their consideration of our revised work and kind words. The authors appreciate the reviewer's recommendations to improve the paper and agree the manuscript findings have been strengthened as a result.

Reviewer 2:

The authors have addressed all my major and minor comments.

In the new Figure 3B, the module colors should be labeled for clarity.

The authors thank the reviewer for their consideration of our revised work and recommendation for improvement to Figure 3B. The following sentence has been added to the caption of Fig. 3 to better clarify the module colors utilized in 3B:

“Eigengene labels, moving left to right in the dendrogram, are: pink, red, tan, navy (ALS-Ox), brown, green, gold (ALS-Glia), gray, maroon (ALS-TD), yellow, blue, salmon, black, and green-yellow.”

Reviewer 3:

The authors have performed a thorough revision of the original manuscripts and incorporated this reviewer's suggestions. Particularly, the question of potential confounders (brain regions, cellular composition heterogeneity, RIN, genetics, etc) has been addressed.

The authors thank the reviewer for their consideration of our revised work and appreciate the reviewer's recommendations to strengthen the manuscript findings.